# Adaptive Divergence Regularized Policy Optimization for Fine-tuning Generative Models

**Jiajun Fan, Tong Wei, Chaoran Cheng, Yuxin Chen, Ge Liu**
University of Illinois Urbana-Champaign
{jiajunf3, twei11, chaoran7, nealchen, geliu}@illinois.edu

## Abstract

Balancing exploration and exploitation during reinforcement learning fine-tuning of generative models presents a critical challenge, as existing approaches rely on fixed divergence regularization that creates an inherent dilemma: strong regularization preserves model capabilities but limits reward optimization, while weak regularization enables greater alignment but risks instability or reward hacking. We introduce Adaptive Divergence Regularized Policy Optimization (ADRPO), which automatically adjusts regularization strength based on advantage estimates—reducing regularization for high-value samples while applying stronger regularization to poor samples, enabling policies to navigate between exploration and aggressive exploitation according to data quality. Our implementation with Wasserstein-2 regularization for flow matching generative models achieves remarkable results on text-to-image generation, achieving better semantic alignment and diversity than offline methods like DPO and online methods with fixed regularization like ORW-CFM-W2. ADRPO enables a 2B parameter SD3 model to surpass much larger models with 4.8B and 12B parameters in attribute binding, semantic consistency, artistic style transfer, and compositional control while maintaining generation diversity. ADRPO generalizes to KL-regularized fine-tuning of both text-only LLMs and multi-modal reasoning models, enhancing existing online RL methods like GRPO while requiring no additional networks or complex architectural changes. In LLM fine-tuning, ADRPO demonstrates an emergent ability to escape local optima through active exploration, while in multi-modal audio reasoning, it outperforms GRPO through superior step-by-step reasoning, enabling a 7B model to outperform substantially larger commercial models including Gemini 2.5 Pro and GPT-4o Audio, offering an effective plug-and-play solution to the exploration-exploitation challenge across diverse generative architectures and modalities.

## 1 Introduction

Reinforcement learning fine-tuning has emerged as a powerful paradigm for aligning generative models with human preferences, driving remarkable improvements in capabilities from text generation to image synthesis [24, 4, 34]. At the core of modern RLHF approaches lies a fundamental challenge: effectively balancing divergence regularization against reward maximization during policy optimization. This balance is critical as it determines whether models retain the beneficial properties of their pre-trained foundation while adapting to better satisfy human preferences [26, 46, 2].

The current standard practice employs divergence regularization with fixed coefficients to constrain policy updates - typically using Kullback-Leibler (KL) [32, 24] or Wasserstein-2 (W2) divergences [1, 14]. However, this approach creates an inherent dilemma that limits performance: strong regularization preserves model capabilities but hampers reward optimization, while weak regularization enables greater reward optimization but risks catastrophic forgetting, mode collapse, or reward

hacking [20, 33]. This trade-off is particularly pronounced in generative models where preserving diversity while improving quality represents a critical balance [3, 15]. Existing approaches like PPO [31], GRPO [32], and DPO [26, 37] employ fixed regularization coefficients that treat all data points equally, regardless of whether the policy should prioritize exploitation (when rewards are reliable) or exploration (when falling into suboptimal solutions). This one-size-fits-all approach fails to adapt to the varying exploration needs across the complex landscapes of generative model policy optimization.

To address these limitations, we propose Adaptive Divergence Regularized Policy Optimization (ADRPO), a novel framework that dynamically adjusts regularization strength using advantage estimates [30, 35]. ADRPO employs advantage signals to fine-tune the exploration-exploitation trade-off: high-advantage samples reduce regularization for aggressive optimization, while low-advantage samples increase it for stability. This sample-level adaptation integrates seamlessly into training, providing an efficient and automated approach. By aligning regularization with sample quality, ADRPO overcomes the shortcomings of prior methods, delivering superior alignment and generative performance across diverse tasks, as demonstrated in our experiments with text-to-image alignment and language model fine-tuning. In summary, our approach makes several important contributions:

1. **General RL Framework with Adaptive Divergence Regularization.** We introduce ADRPO as a general-purpose framework that dynamically adjusts regularization based on advantage estimates, integrating with existing RL fine-tuning methods without architectural changes. Our proposed methods address the exploration-exploitation dilemma while preventing reward hacking and model collapse.

2. **Superior Text-to-Image Alignment with Smaller Model.** We first propose a novel online RL method based on ADRPO, combining advantage-based policy optimization and adaptive W2 regularization for fine-tuning flow matching models. Our experiments of fine-tuning SD3 demonstrate ADRPO's dominant Pareto frontier in the reward-diversity trade-off and reward-divergence trade-off compared to DPO [26] and fixed-regularization approaches [14] (See Figs. 2 and 3). Notably, our 2B parameter model outperforms larger 4.8B [38] and 12B [43] parameter models across attribute binding, compositional control, and semantic consistency (See Figs. 1 and Tab. 1).

3. **Emergent Exploration in LLMs.** We also apply our ADRPO to improve GRPO [32] for online fine-tuning of LLMs (See Figs. 4). ADRPO not only improves alignment but exhibits an emergent ability to escape local optima by actively increasing exploration when needed—a capability absent in fixed-regularization methods like GRPO.

4. **Cross-Domain Applicability.** ADRPO provides a unified solution across continuous (flow matching with W2 regularization), discrete (LLMs with KL divergence), and multi-modal reasoning generative paradigms (audio reasoning models), offering immediate practical benefits with minimal computational overhead (See Sec. 4.6 and App. D.3).

## 2 Related Work

**RL Fine-tuning for LLMs.** Reinforcement learning has become the dominant approach for aligning large language models with human preferences. Pioneering work by [4] established the RLHF framework, which was later scaled by [24] to create models that better follow human instructions. The algorithmic landscape has evolved from PPO [31] to more efficient alternatives like GRPO [32] and offline approaches like DPO [26]. These methods have significantly improved the reasoning capabilities of models like DeepSeek-R1 [16] and improved their instruction-following abilities. Despite their success, these approaches typically rely on fixed regularization parameters that treat all samples equally, regardless of whether they represent promising directions for optimization or uncertainty-laden regions requiring more conservative updates (See Fig. 4).

**RL Fine-tuning for Flow Matching Models.** While RL fine-tuning is established for language models, its application to flow matching (FM) models [23] presents unique exploration-exploitation trade-off challenges due to their continuous-time nature and ODE-based sampling. Recent approaches like Online Reward-Weighted Fine-Tuning with Wasserstein regularization [14] and offline methods like diffusion-DPO [37] have made progress, but remain limited by fixed regularization schemes that cannot adapt to sample-specific characteristics. This fundamental limitation restricts their ability to optimally balance the critical exploration-exploitation trade-off necessary for effective fine-tuning of state-of-the-art image generation models like SD3 [13] (See Tab. 1 and Figs. 2).

**Divergence Regularization in RL Fine-tuning.** Divergence regularization plays a crucial role in RL fine-tuning by preventing the policy from deviating too far from the initial model, thus preserving desirable properties while allowing for improvement. For language models, KL divergence serves as the standard metric in methods like PPO [31], GRPO [32], and DPO [26], while flow models benefit from Wasserstein distances [14] that better handle continuous distributions. Despite their importance, existing approaches typically employ fixed regularization coefficients that fail to handle the varying significance of regularization across different samples and different learning stages. This limitation can lead to suboptimal trade-offs between preserving model capabilities and maximizing rewards (e.g., GRPO in Fig. 4), risking model collapse [33, 17] or insufficient improvement. Our work addresses this gap through adaptive regularization based on advantage estimates, providing a novel approach to dynamically balancing exploration and exploitation during training.

## 3 Method

### 3.1 Problem Formulation

In this paper, we address the challenges of fine-tuning pre-trained generative models through online RL to improve their alignment with human preferences [4, 24]. Given a pre-trained reference policy $\pi_{\text{ref}}$ and its fine-tuned counterpart $\pi_\theta$ parameterized by $\theta$, our objective is to maximize the expected user-defined reward $\mathbb{E}_{x \sim \pi_\theta}[R(x, c)]$, where $R(x, c)$ quantifies human preference for generation $x$ conditioned on context $c \sim p(c)$ (e.g., CLIP Score [25] for T2I tasks). This context may be a text prompt in LLMs [41, 42, 11] or an image description in text-to-image (T2I) models [13, 38, 43]. The standard approach in RL fine-tuning formulates this as a constrained optimization problem:

$$J(\theta) = \mathbb{E}_{x \sim \pi_\theta, c \sim p(c)}[R(x, c)] - \beta \cdot D(\pi_\theta, \pi_{\text{ref}}) \tag{1}$$

Here, $p(c)$ is the sample distribution of prompts (e.g., uniform sampling in our paper), $D(\pi_\theta, \pi_{\text{ref}})$ represents a divergence measure between the fine-tuned and reference policies—typically Kullback-Leibler divergence (KL) for discrete generative models [31, 32, 26] or Wasserstein distance for continuous distributions [36, 1, 14]. The coefficient $\beta$ controls the trade-off between reward optimization and preservation of the pre-trained model's capabilities (e.g., diversity).

### 3.2 Adaptive Divergence Regularized Policy Optimization

Recent approaches to online RL fine-tuning of generative models have explored different divergence measures, including W2 regularization in flow matching models [14] and KL divergence in LLMs [32, 24]. However, these methods still rely on fixed regularization schemes that treat all samples equally, regardless of their potential for reward improvement or risk of degradation. This fundamental challenge of adaptive regularization—dynamically balancing exploration and exploitation (See Figs. 3 and 4) at the individual sample level—remains largely unaddressed in the literature.

#### 3.2.1 Conventional RL Fine-tuning Approaches

The conventional RL objective in Equation (1) can be rewritten as a combination of two loss terms:

$$\mathcal{L}_{\text{RLHF}}(\theta) = \mathcal{L}_{\text{RL}}(\theta) + \beta \cdot \mathcal{L}_{\text{D}}(\theta) \tag{2}$$

where $\mathcal{L}_{\text{RL}}(\theta)$ is the policy optimization term such as policy gradient [31] or reward-weighting [14] and $\mathcal{L}_{\text{D}}(\theta) = D(\pi_\theta, \pi_{\text{ref}})$ is the divergence regularization term, such as KL divergence in LLMs [32] or W2 divergence in flow matching models [14]. In practice, this formulation has been instantiated in various ways. For example, Group Relative Policy Optimization (GRPO) [32] employs a KL-regularized policy gradient objective for LLMs:

$$\mathcal{L}_{\text{GRPO}}(\theta) = \mathcal{L}_{\text{PG}}(\theta) + \beta \cdot D_{\text{KL}}(\pi_\theta \| \pi_{\text{ref}}) \tag{3}$$

where $\mathcal{L}_{\text{PG}}(\theta)$ represents a clipped policy gradient loss based on group-level advantage estimation. Similarly, ORW-CFM-W2 [14] applies a W2 regularization term for flow matching models:

$$\mathcal{L}_{\text{ORW-CFM-W2}}(\theta) = \mathcal{L}_{\text{ORW}} + \beta \cdot \mathbb{E}_{c,t,x_t}[|\mathbf{v}_\theta(x_t, t, c) - \mathbf{v}_{\text{ref}}(x_t, t, c)|^2]$$

where $\mathcal{L}_{\text{ORW}} = \mathbb{E}_{c,x_1,t,x_t}[\omega(x_1, c) * |\mathbf{v}_\theta(x_t, t, c) - \mathbf{u}_t|^2]$ is the reward weighted loss and $\mathbf{v}_\theta$ and $\mathbf{v}_{\text{ref}}$ are the velocity fields of the fine-tuned and reference policies, respectively.

Critically, in all these approaches, the regularization strength $\beta$ remains constant across all samples and training steps, failing to adapt to the varying quality of generated samples.

### 3.2.2 Our Approach: Advantage-Based Adaptive Regularization

We introduce Adaptive Divergence Regularized Policy Optimization (ADRPO), a principled framework that dynamically adjusts regularization strength based on the estimated advantages of individual samples. The key insight in ADRPO is that the regularization coefficient should not be static, but should vary inversely with the sample's estimated advantage. Formally, we propose:

$$\mathcal{L}_{\text{ADRPO}}(\theta) = \mathcal{L}_{\text{RL}}(\theta) + (\beta_0 - A) \cdot \mathcal{L}_{\text{D}}(\theta) \tag{4}$$

where $A$ is an advantage estimate for the current sample and $\beta_0$ is a baseline regularization coefficient. This formulation creates an adaptive regularization coefficient $\beta_{\text{tot}} = \beta_0 - A$ that adapts based on the quality of each sample. This adaptive mechanism creates a natural balance: **1) Exploitation:** in regions where the policy generates high-quality samples (high advantage), ADRPO allows for efficient exploitation by reducing divergence penalties; **2) Exploration:** in uncertain or low-quality regions (low advantage), it enforces stronger regularization to maintain stability and preserve the model's original capabilities (See Figs. 3 and 4).

Based on Equ. (4), our ADRPO can be seamlessly integrated with various existing RL fine-tuning methods. For instance, when applied to GRPO for large language models (LLMs), the objective becomes $\mathcal{L}_{\text{ADRPO-GRPO}}(\theta) = \mathcal{L}_{\text{PG}}(\theta) + (\beta_0 - A_{\text{GRPO}}) \cdot D_{\text{KL}}(\pi_\theta \| \pi_{\text{ref}})$, where $A_{\text{GRPO}}$ is the advantage estimate from GRPO's group-based estimation procedure [32, 16].

## 3.3 ADRPO for Flow Matching Generative Models

We now demonstrate how our ADRPO framework can be effectively applied to fine-tuning flow matching models [23, 36], particularly focusing on text-to-image generation models like SD3 [13].

### 3.3.1 Flow Matching Preliminaries

Flow matching (FM) models define a continuous-time transformation that maps a simple prior distribution $p(x_0)$ (e.g., Gaussian) to a complex target distribution via a probability flow $p_t$. An FM model learns a velocity field $\mathbf{v}_\theta(x_t, t, c)$ that approximates the true velocity field $\mathbf{u}_t(x_t|c)$. However, since $\mathbf{u}_t(x_t|c)$ is often intractable [23], Conditional Flow Matching (CFM) [36] proposes an equivalent yet tractable objective by conditioning the flow on target samples $x_1$ while learning a conditional target velocity field (e.g., $\mathbf{u}_t(x_t|x_1, c) = x_1 - x_0$ for linear interpolation path [23]):

$$\mathcal{L}_{\text{CFM}}(\theta) = \mathbb{E}_{c \sim p(c), t \sim U(0,1), x_1 \sim p_{\text{data}}(x|c), x_t \sim p_t(x_t|x_1, c)}[|\mathbf{v}_\theta(x_t, t, c) - \mathbf{u}_t(x_t|x_1, c)|^2] \tag{5}$$

Given a pre-trained reference model like SD3 [13], flow matching fine-tuning aims to align generations with human preferences while preserving generative diversity. Traditional approaches, including supervised fine-tuning and offline RL methods like DPO [26, 37], sample target states $x_1 \sim p_{\text{data}}(x|c)$ from a fixed human-curated dataset—a stable but limiting approach that restricts exploration of potentially better policy regions (See Figs. 2 and 3 ). In contrast, our proposed ADRPO framework embraces an online RL paradigm, sampling target states from the fine-tuned policy itself: $x_1 \sim p_\theta^{n-1}(x|c)$, with $p_\theta^{n-1}$ representing the policy at the previous iteration. This online sampling strategy enables the model to continuously improve upon its own generations and explore the policy space more effectively but is prone to collapse [14], while our adaptive regularization mechanism specifically addresses the inherent instability and exploration-exploitation dilemma in online RL fine-tuning.

### 3.3.2 ADRPO with Wasserstein Regularization

A key observation across RL fine-tuning methods [32, 14, 24] is that effective policy optimization requires differentially weighting samples based on quality (e.g., upweighting probabilities of high-reward samples while downweighting poor ones). While traditional RL methods scale updates by advantage estimates [31, 30], this principle—strengthening high-quality trajectories while weakening low-quality ones based on advantage estimates [31, 35]—hasn't been fully leveraged in flow matching fine-tuning. Current approaches like reward-weighted flow matching [14]

can only down-weight poor samples without actively discouraging them, significantly reducing efficiency in high-dimensional spaces (e.g., image generation) where undesirable regions vastly outnumber desirable ones. We address this limitation by introducing advantage-based policy optimization for flow matching models, creating bidirectional learning signals through advantage estimates rather than non-negative reward weights to both enhance high-quality generations and actively suppress poor ones. Specifically, we propose an advantage-weighted flow matching objective: $\mathcal{L}_{\text{RL}}(\theta) = \mathbb{E}_{c\sim p(c),t\sim U(0,1),x_1\sim p_\theta^{n-1},x_t\sim p_t(x_t|x_1,c)}[A(x_1,c) \cdot |\mathbf{v}_\theta(x_t,t,c) - \mathbf{u}_t|^2]$, where $p_\theta^{n-1}$ is the current fine-tuned policy and $A(x_1,c)$ represents the estimated advantage for sample $x_1$ given context $c$, $\mathbf{v}_\theta(x_t,t,c)$ is the learned velocity field, $x_t = (1-t)x_0 + tx_1$, and $\mathbf{u}_t = x_1 - x_0$ is the target velocity for the straight-line interpolation in FM models [23].

This formulation creates a fundamentally different learning dynamic compared to reward-weighting approaches. For samples with positive advantage ($A > 0$), the objective encourages matching the target velocity field, strengthening high-quality generations. Conversely, for samples with negative advantage ($A < 0$), the sign inversion reverses the gradient direction, actively pushing the model away from poor generations rather than merely down-weighting them. Meanwhile, average-quality samples (where $A \approx 0$) contribute minimally to the gradient, naturally focusing computational resources on the most informative examples and facilitating efficient convergence (See Fig. 3).

**Advantage Estimation.** For FM models, we compute the advantage as the difference between the reward of a sample and the expected reward under the current policy as $A(x_1,c) = R(x_1,c) - V(c)$, where $R(x_1,c)$ is the human preference reward for the generated sample $x_1$ given context $c$, and $V(c)$ is a baseline value function estimated as the average reward over a batch of samples for the same context, which is computationally efficient.

**Adaptive Regularization.** Based on Equ. (4), we further propose to dynamically adjust the regularization strength based on the same advantage estimates. This creates a unified framework where the exploration-exploitation balance is automatically modulated at the individual sample level:

$$\mathcal{L}_{\text{ADRPO-FM}}(\theta) = \mathbb{E}_{c\sim p(c),t\sim U(0,1),x_1\sim p_\theta^{n-1},x_t\sim p_t(x_t|x_1,c)}[A(x_1,c) \cdot |\mathbf{v}_\theta(x_t,t,c) - \mathbf{u}_t|^2]$$
$$+(\beta_0 - A(x_1,c)) \cdot \mathbb{E}_{c,t,x_t}[|\mathbf{v}_\theta(x_t,t,c) - \mathbf{v}_{\text{ref}}(x_t,t,c)|^2] \tag{6}$$

The adaptive regularization coefficient $\beta_{\text{tot}} = \beta_0 - A(x_1,c)$ establishes a dynamic adaptation mechanism responsive to sample quality. For high-advantage samples ($A > 0$), regularization decreases proportionally. For low-advantage samples ($A < 0$), regularization strengthens proportionally, constraining updates to maintain proximity to the reference model. This bidirectional adaptation fundamentally transforms the exploration-exploitation landscape (Fig. 3), replacing fixed regularization with sample-wise W2 regularization that continuously adapt to the evolving policies.

**Stabilization, Efficient Learning.** To ensure stable training with our adaptive advantage-based approach, we use advantage clipping that constrains advantages to a reasonable range $[A_{\min}, A_{\max}]$ as $A_{\text{clipped}}(x_1,c) = \text{clip}(A(x_1,c), A_{\min}, A_{\max})$. We also use LoRA [18] for efficient learning.

### 3.4 ADRPO for Fine-tuning LLMs

Applying our ADRPO framework to Large Language Models (LLMs) can address the limitation of static regularization in conventional online RL methods by dynamically controlling the penalty for deviating from the pre-trained policy based on sample advantage. High-advantage responses indicate promising directions warranting reduced regularization to encourage policy optimization, while low-advantage responses signal areas to avoid, requiring increased regularization to maintain proximity to the reliable pre-trained model and prevent undesirable outputs or instability. We integrate this principle with GRPO [32], modifying its objective by making the KL divergence regularization strength dependent on the advantage estimate ($A_{\text{GRPO}}$) for each sample. The objective becomes:

$$\mathcal{L}_{\text{ADRPO-GRPO}}(\theta) = \mathcal{L}_{\text{PG}}(\theta) + (\beta_0 - A_{\text{GRPO}}) \cdot D_{\text{KL}}(\pi_\theta \| \pi_{\text{ref}}) \tag{7}$$

Here, $\mathcal{L}_{\text{PG}}(\theta)$ is the clipped policy gradient term [32] (i.e., $-\min(A*\text{ratio}, A*\text{clip}(\text{ratio}, 1-\epsilon, 1+\epsilon))$ and ratio $= \frac{\pi_\theta}{\pi_{\theta_{\text{old}}}}$), $D_{\text{KL}}$ is the KL divergence, and $\beta_0$ is a baseline regularization. The term $(\beta_0 - A_{\text{GRPO}})$ acts as an adaptive coefficient, decreasing for good samples ($A_{\text{GRPO}} > 0$) to promote

Table 1: Comparison of text-to-image generation methods across different evaluation metrics. Best scores are highlighted in blue , second-best in green . We report standard errors estimated over 3 random seeds. ClipDiversity measures the mean pairwise distance of CLIP embeddings [25, 14].

| Method | Task Metrics | | Image Quality | Human Preference | | |
|---|---|---|---|---|---|---|
| | ClipScore↑ [25] | ClipDiversity↑ [25] | Aesthetic↑ [39] | BLIPScore↑ [12] | ImageReward↑ [39] | PicScore↑ [21] |
| *Base Model* | | | | | | |
| SD3 (2B) [13] | 29.27±0.42 | 5.08±0.52 | 5.53±0.09 | 0.501±0.007 | 0.97±0.13 | 20.81±0.09 |
| *Other Flow Matching Models* | | | | | | |
| FLUX.1-Dev (12B) [43] | 31.72±0.48 | 4.29±0.42 | 5.95±0.05 | 0.492±0.004 | 1.11±0.10 | 21.83±0.11 |
| SANA-1.5 (4.8B) [38] | 32.18±0.36 | 4.31±0.50 | 5.89±0.12 | 0.526±0.006 | 1.45±0.08 | 21.85±0.15 |
| *SD3 Fine-tuning Methods* | | | | | | |
| SD3+RAFT [9] | 29.35±0.27 | 1.85±0.19 | 4.54±0.04 | 0.512±0.001 | 0.22±0.08 | 19.21±0.02 |
| SD3+DPO [37] | 31.30±0.52 | 4.78±0.46 | 5.82±0.05 | 0.509±0.005 | 1.48±0.10 | 21.31±0.10 |
| SD3+ORW-CFM-W2 [14] | 31.42±0.39 | 3.86±0.37 | 5.29±0.05 | 0.542±0.006 | 1.22±0.10 | 20.97±0.11 |
| SD3+ADRPO (Ours) | 32.97±0.46 | 5.13±0.47 | 6.27±0.06 | 0.567±0.004 | 1.61±0.05 | 22.78±0.15 |

exploitation and increasing for poor samples ($A_{\text{GRPO}} < 0$) to enforce conservative exploration, allowing ADRPO-GRPO to achieve a better exploration-exploitation trade-off (See Fig. 4).

# 4 Experiment

## 4.1 Experimental Setup

For our experiments, we evaluated ADRPO across three distinct domains: flow matching models, LLMs, and multi-modal reasoning models. **Fine-tuning Flow Matching Models.** We implemented ADRPO on SD3 (2B parameters) using prompts from DrawBench [28] testing color attribute binding, compositional reasoning, object counting, spatial relationships, and text rendering, as well as artistic style transfer prompts from RAFT [9]. Our method employed the advantage-based ADRPO loss from Equation (6) with $\beta_0 = 1$ and $A_{max} = 1, A_{min} = -1$, using CLIP score as rewards [25]. We compared against offline methods like DPO [37], online approaches like ORW-CFM-W2 [14], and substantially larger models including FLUX.1 Dev (12B) [43] and SANA-1.5 (4.8B) [38]. **Fine-tuning LLMs.** We fine-tuned Qwen2 [41] and Qwen3 [42] using RM-Gemma-2B [27, 9] as the reward model on RLHFlow/test_generation_2k dataset [9]. ADRPO was integrated with GRPO using KL-divergence regularization (Equation (7)) with $\beta_0 = 0.04, A_{min} = -0.04, A_{max} = 0.04$, compared against standard GRPO with fixed $\beta = 0.04$ [32]. **Fine-tuning Multi-Modal Reasoning Models.** We fine-tuned Qwen2.5-Omni-7B [40] on the AVQA dataset [44] using verifiable and format rewards, evaluated on the MMAU benchmark [29]. We used $\beta_0 = 0.04, A_{min} = -0.04, A_{max} = 0.04$, comparing against GRPO baseline and commercial models including Gemini 2.5 Pro [5] and GPT-4o Audio [19]. All experiments employed advantage clipping for training stability and set $\beta_0$ equal to the fixed $\beta$ in baseline methods for fair comparison. See App. B and C for complete details.

## 4.2 Main Results

Tab. 1 presents a comprehensive evaluation of text-to-image generation methods, demonstrating that our proposed ADRPO establishes a superior Pareto frontier in all metrics and achieves the best reward-diversity trade-off. Unlike competing approaches such as DPO and ORW-CFM-W2 that make significant compromises—improving semantic alignment at the cost of diversity or vice versa—ADRPO achieves state-of-the-art performance in both dimensions simultaneously through its dynamic regularization mechanism in Equ. (6). Our adaptive approach intelligently modulates regularization strength based on sample-specific advantage estimates, enabling aggressive exploitation in high-reward regions while maintaining exploration elsewhere (See Figs. 3 and 4). Perhaps most remarkably, our method enables a relatively modest 2B parameter SD3 model to outperform substantially larger models including FLUX.1-Dev (12B) [43] and SANA-1.5 (4.8B) [38] across all evaluation metrics, particularly in human preferences. This quantitative superiority is visually evident in our qualitative results in Figs. 1 and 2 where ADRPO-generated images demonstrate exceptional attribute binding, spatial understanding, text rendering, and artistic style transfer capabilities that

even larger models struggle to match. Together, these findings suggest that adaptive regularization offers a more efficient path to performance improvement than simply scaling model parameters.

## 4.3 Qualitative Analysis

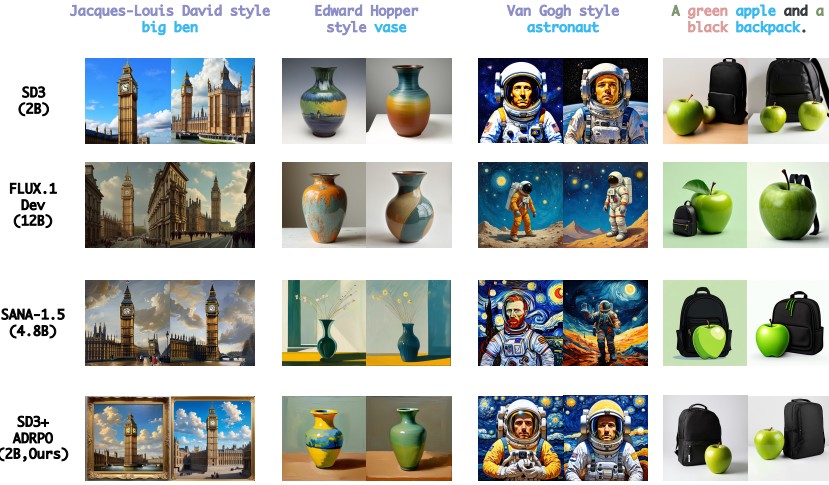

Figure 1: **Qualitative Comparison with Large FM Generative Models.** Our ADRPO demonstrates superior performance in Artistic Style Rendering, Attribute Binding, Coloring and Counting.

**Comparison with SOTA Large FM Models.** Fig. 1 shows our ADRPO fine-tuned SD3 model (2B parameters) significantly outperforming much larger models like FLUX.1 Dev (12B) and SANA-1.5 (4.8B). This challenges the conventional wisdom that parameter scaling is the primary path to performance improvements. Our method excels in areas where larger models struggle: for artistic style transfer ("Jacques-Louis David style big ben"), complex compositions ("Van Gogh style astronaut"), and attribute binding ("green apple and black backpack"), ADRPO maintains both style accuracy and compositional integrity while larger models introduce visual artifacts despite their 2-6× parameter counts. These results demonstrate that adaptive regularization can enable smaller models to match or exceed much larger models' capabilities. See App. D for more results.

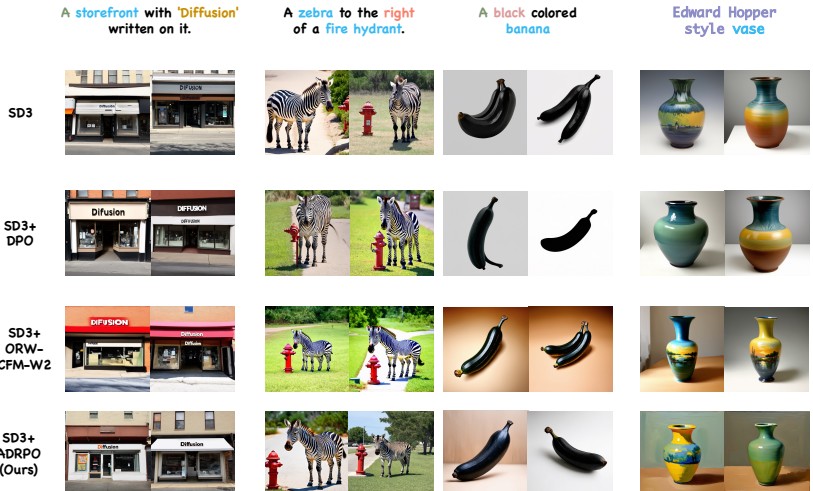

Figure 2: **Qualitative Comparison with Other RL Fine-tuning Methods.** Our ADRPO demonstrates superior performance in Artistic Style Rendering, Text Rendering, Attribute Binding, Coloring, Counting and Position. We use a similar DPO method as described in [8] to fine-tune SD3 models.

**Comparison with other RL Fine-tuning Methods.** Fig. 2 demonstrates ADRPO's clear superiority over existing reinforcement learning fine-tuning approaches. While DPO [37] preserves diversity at the cost of semantic alignment and ORW-CFM-W2 [14] improves alignment but sacrifices diversity, ADRPO achieves excellence in both dimensions through advantage-guided regularization. This is evident across text rendering ("Diffusion" storefront), attribute binding (zebra positioning), coloring (black banana), and style transfer tasks, where our method consistently delivers superior compositional accuracy. By dynamically modulating regularization strength—increasing constraints for uncertain samples while allowing greater divergence for reliable ones—ADRPO effectively resolves the exploration-exploitation dilemma that static approaches cannot address.

## 4.4 Visualizing Exploration-Exploitation Trade-off in Policy Optimization

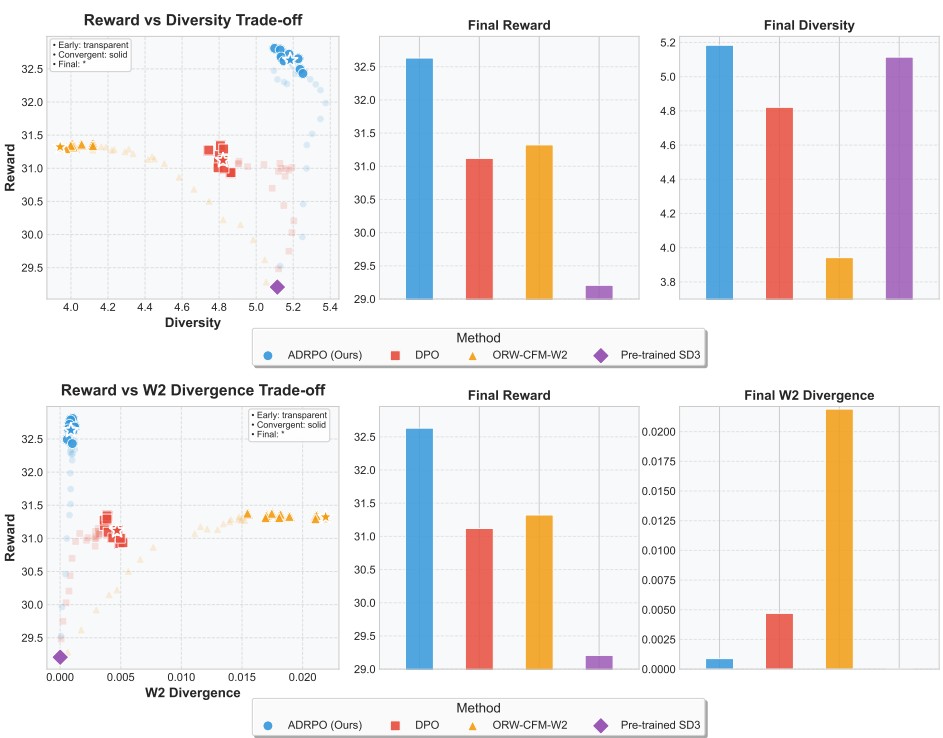

Figure 3: **Reward-Diversity/Divergence Trade-off.** Left: policy optimization trajectories (using a same seed) of different methods throughout training, with transparency indicating progression from early (transparent) to convergent (solid) to final (star) checkpoints. Each point is a learned policy from different iterations. Center and right: final reward and diversity/divergence across methods.

**Reward Hacking Mitigation.** Figs. 3 reveals distinct vulnerability patterns to reward hacking across methods. While DPO maintains moderate diversity but plateaus in reward optimization, ORW-CFM-W2 aggressively pursues reward optimization but exhibits significant diversity collapse (right panel), resulting in template-like generations (See Fig. 2). Our ADRPO, through advantage-guided regularization, achieves the highest reward without sacrificing diversity—a combination neither competing method attains. This translates to superior generations with precise attribute binding and high visual quality while maintaining creative flexibility. See App. D for whole learning curves.

**Exploration-Exploitation Balance and Divergence Control.** The trajectory visualization in Figs. 3 (left) captures each method's navigation of the exploration-exploitation landscape. The bottom plots further illustrate ADRPO's advantage in maintaining minimal W2 divergence while maximizing reward. While DPO makes modest improvements before plateauing and ORW-CFM-W2 follows an exploitation path that compromises diversity, ADRPO consistently expands the Pareto frontier. This superior balancing act stems from our adaptive mechanism making sample-specific regularization decisions, effectively resolving the dilemma that fixed-coefficient methods cannot address.

## 4.5  Application to Fine-tuning LLMs

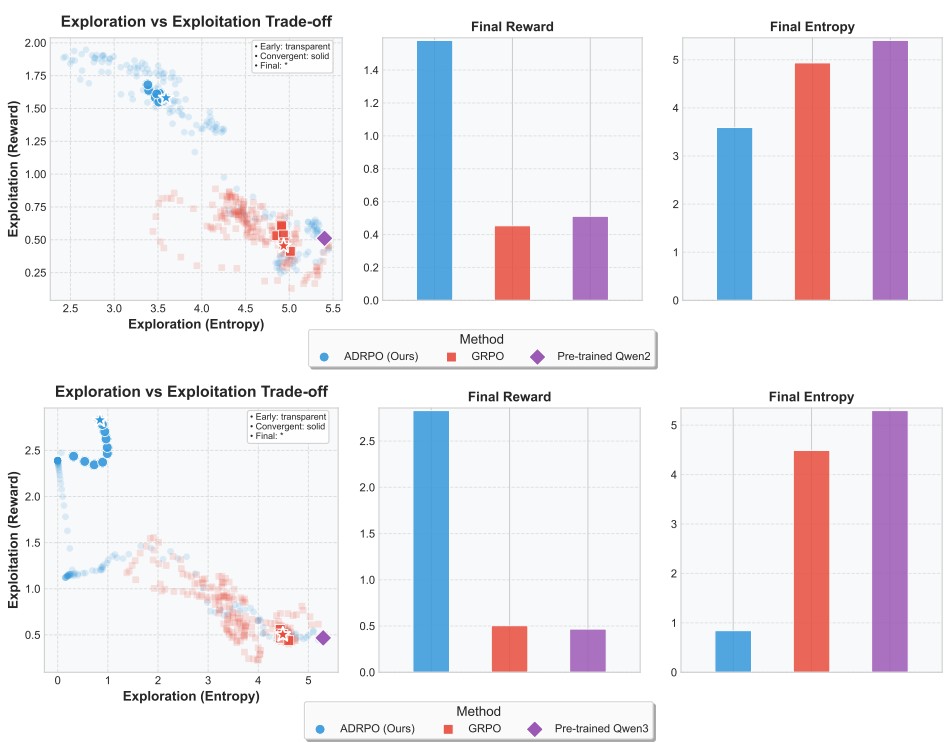

Figure 4: Ablation Studies and Exploration-Exploitation Trade-off in Fine-tuning LLMs. Left: policy optimization trajectories in reward-entropy space for ADRPO and GRPO (i.e., fixed $\beta$ as [32]) across Qwen2 (0.5B) [41] (top) and Qwen3 (0.6B) [42] (bottom) models, with transparency indicating progression from early to final checkpoints. Center and right: Final performance of different methods.

To demonstrate ADRPO's versatility beyond FM models, we applied our approach to LLM fine-tuning with the Qwen2 [41] and Qwen3 [42] model families using RM-Gemma-2B [9, 10, 27] as the reward model, as shown in Fig. 4. All methods are evaluated in RLHFlow/test_generation_2k dataset [9].

**Superior Exploration-Exploitation Control.** The policy optimization trajectories (left panels) in Fig. 4 reveal distinct patterns between methods. While GRPO maintains high entropy throughout training but struggles to find high-reward regions—showing substantial horizontal movement with limited vertical improvement—ADRPO implements a strategic exploration pattern that efficiently navigates the exploration-exploitation landscape. For Qwen3 (bottom left), ADRPO exhibits a remarkable ability to first explore lower entropy regions, then actively increase entropy to escape local optima, before converging to a final checkpoint with 5× higher reward than GRPO.

**Preventing Model Collapse.** ADRPO demonstrates superior resistance to model collapse during extended training. GRPO's performance tends to plateau or deteriorate as training progresses (also see Fig. D.2.1), with later checkpoints (darker red points) often showing lower rewards than earlier ones—a common failure mode of fixed-regularization methods. In contrast, ADRPO shows consistent improvement throughout training by dynamically adjusting regularization strength based on advantage estimates, eliminating the need for careful early stopping to prevent performance regression.

**Cross-Architecture Generalizability.** The consistent superior performance across both flow matching models and different LLM architectures confirms that ADRPO addresses fundamental limitations in reinforcement learning fine-tuning. By adapting regularization strength to sample-specific advantage estimates, our method provides a generalizable solution to the exploration-exploitation dilemma that effectively transfers between domains. See App. D for whole learning curves.

Table 2: Multi-modal audio reasoning results on MMAU benchmark [29].

| Method | Sound (%) | Music (%) | Speech (%) | Total (%) |
|---|---|---|---|---|
| Qwen2.5-Omni (base) | 72.37 | 64.37 | 69.07 | 68.6 |
| GRPO | 77.18 | 70.66 | 74.77 | 74.2 |
| Gemini 2.5 Pro | 75.08 | 68.26 | 71.47 | 71.6 |
| GPT-4o Audio | 64.56 | 56.29 | 66.67 | 62.5 |
| **ADRPO (Ours)** | **81.98** | **70.06** | **75.98** | **76.0** |

Table 3: Ablation on advantage clipping ranges.

| Clipping Range | $A_{\min}$ | $A_{\max}$ | Sound (%) | Music (%) | Speech (%) | Total (%) |
|---|---|---|---|---|---|---|
| $1 \times \beta_0$ (recommended) | -0.04 | 0.04 | 81.98 | 70.06 | 75.98 | **76.0** |
| $2 \times \beta_0$ | -0.08 | 0.08 | 84.08 | 69.46 | 73.57 | 75.7 |
| $0.5 \times \beta_0$ | -0.02 | 0.02 | 82.58 | 71.26 | 74.47 | 76.1 |
| GRPO (baseline) | - | - | 77.18 | 70.66 | 74.77 | 74.2 |

## 4.6 Extension to Multi-Modal Audio Reasoning

To validate ADRPO's generalizability beyond text-to-image generation and text-only LLMs, we fine-tuned Qwen2.5-Omni-7B [40] on audio reasoning tasks using the AVQA dataset [44], evaluated on the MMAU benchmark [29]. This domain requires temporal sequence understanding and acoustic pattern recognition—capabilities distinct from text or visual generation.

Tab. 2 presents our results comparing ADRPO against the base model, GRPO baseline, and larger commercial models. Our method achieves 76.0% accuracy, outperforming all baselines and enabling a 7B model to surpass substantially larger systems including Gemini 2.5 Pro [5] and GPT-4o Audio [19] through more effective exploration-exploitation balance.

**Hyperparameter Robustness.** To assess ADRPO's practical applicability in new domains, we conducted ablation studies on advantage clipping ranges. Tab. 3 shows that all ADRPO configurations substantially outperform GRPO with stable performance across settings. The recommended $1 \times \beta_0$ achieves the best overall balance, while $2 \times \beta_0$ shows the trade-off between aggressive exploitation (higher sound accuracy) and generalization (lower music/speech accuracy). Performance variations across clipping ranges remain under 0.4%, confirming robustness.

These results, combined with our text-to-image experiments (Sec. 4) and LLM fine-tuning (Sec. 4.5), demonstrate ADRPO's effectiveness across continuous generation, discrete sequences, and multi-modal reasoning. Qualitative analysis reveals that ADRPO enables superior step-by-step reasoning on complex temporal and contextual tasks compared to GRPO's premature convergence. See App. D.3 for detailed setup and qualitative examples.

## 5 Conclusion

The exploration-exploitation dilemma represents a critical challenge in generative model RL fine-tuning that fixed regularization approaches fail to address. To tackle this, we propose Adaptive Divergence Regularized Policy Optimization (ADRPO), which dynamically adjusts regularization strength based on sample-specific advantage estimates—reducing constraints for high-value samples while strengthening them for poor ones. Our experiments demonstrate ADRPO's effectiveness across diverse domains and modalities: in text-to-image generation, it outperforms other methods in alignment, quality, and diversity, enabling our 2B parameter SD3 model to surpass much larger models (4.8B and 12B) in various tasks; in LLM fine-tuning, it exhibits an emergent ability to escape local optima by actively increasing exploration; in multi-modal audio reasoning, it outperforms GRPO through superior step-by-step reasoning, enabling a 7B model to outperform substantially larger commercial models including Gemini 2.5 Pro and GPT-4o Audio. ADRPO establishes a superior Pareto frontier in the reward-diversity trade-off across continuous generation, discrete sequences, and multi-modal reasoning tasks, confirming that sample-adaptive regularization offers a plug-and-play solution that generalizes across generative architectures with minimal computational overhead. See App. A for more discussion.

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

# A Discussion

In reinforcement learning fine-tuning of generative models, the exploration-exploitation trade-off represents a critical challenge: too much exploitation leads to reward hacking and diversity collapse, while excessive exploration prevents effective alignment. This dilemma is particularly pronounced in online RL fine-tuning where models continuously learn from their own generations. Existing methods rely on fixed regularization coefficients that treat all samples equally, regardless of their reward potential or uncertainty, creating an inherent tension between preserving model capabilities and optimizing for reward.

> **Key Insight 1:** ADRPO solves the fundamental exploration-exploitation dilemma in generative model fine-tuning through a principled adaptive mechanism where regularization strength automatically scales with sample-specific advantage estimates.

In this paper, we introduced Adaptive Divergence Regularized Policy Optimization (ADRPO), a novel approach that fundamentally reimagines how divergence regularization is applied during generative model fine-tuning. Unlike existing approaches that employ fixed regularization coefficients, ADRPO dynamically modulates regularization strength based on sample-specific advantage estimates, effectively turning the static trade-off parameter into an adaptive function. Through comprehensive experimentation, we have demonstrated that this simple yet powerful modification successfully overcomes the inherent limitations of fixed-regularization methods across both text-to-image generation and language model domains, establishing a new paradigm for reinforcement learning fine-tuning of generative models.

> **Key Finding 1:** ADRPO establishes a dominant Pareto frontier in the reward-diversity trade-off, achieving state-of-the-art performance in both dimensions simultaneously where previous methods could only optimize one at the expense of the other (Fig. 3).

Our experimental results reveal critical insights about the nature of reinforcement learning fine-tuning. The reward-diversity trade-off, long considered an unavoidable compromise in generative model alignment, can be effectively navigated through our adaptive regularization framework. As shown in Fig. 3 (left), DPO preserves moderate diversity but plateaus in reward optimization, while ORW-CFM-W2 improves alignment but suffers significant diversity collapse. In contrast, ADRPO achieves dominant performance in both dimensions simultaneously (Fig. 3, center and right panels), establishing a new state-of-the-art that was previously considered unattainable. This is achieved through our bidirectional adaptation mechanism, where regularization strength decreases for high-advantage samples to enable aggressive optimization while increasing for low-advantage samples to maintain stability and diversity.

> **Key Finding 2:** ADRPO enables remarkable parameter efficiency, allowing a 2B parameter model to consistently outperform substantially larger models (4.8B and 12B), demonstrating that optimization strategy can be more consequential than model scale (Tab. 1).

The parameter efficiency enabled by our approach challenges fundamental assumptions about scaling laws in generative AI. As demonstrated in Tab. 1, our 2B parameter SD3 model fine-tuned with ADRPO consistently outperformed substantially larger models including FLUX.1-Dev (12B) and SANA-1.5 (4.8B) across all evaluation metrics—from ClipScore to human preference metrics like ImageReward and PicScore. This finding suggests that optimization strategy can be more consequential than raw parameter count for generative quality, with profound implications for both research focus and practical deployment. By enabling smaller models to match or exceed the capabilities of models 2-6× their size, ADRPO offers a path to personal access to high-quality generative AI while significantly reducing computational and environmental costs. The qualitative results in Fig. 1 further support this finding, showing superior attribute binding and style transfer capabilities compared to larger models.

> **Key Finding 3:** ADRPO exhibits an emergent ability to intelligently navigate the exploration-exploitation landscape, actively increasing exploration to escape local optima—a sophisticated capability that emerges naturally from our advantage-guided mechanism (Fig. 4).

Perhaps the most surprising property of ADRPO is its emergent ability to escape local optima through strategic exploration. As visualized in Fig. 4, our LLM experiments with Qwen3 revealed that

ADRPO implements a sophisticated optimization trajectory absent in fixed-regularization methods. While GRPO maintained high entropy throughout training but struggled to find high-reward regions (Fig. 4, bottom left, red points), ADRPO exhibited a remarkable three-phase pattern: first exploring lower entropy regions, then actively increasing entropy to escape local optima, before finally converging to a high-reward solution (Fig. 4, bottom left, blue trajectory). This behavior—reminiscent of sophisticated simulated annealing schedules—emerged organically from our advantage-guided adaptive mechanism without explicit programming, resulting in final checkpoints with 5× higher reward than GRPO (Fig. 4, bottom center). This finding suggests that advantage-guided regularization may unlock entirely new regions of policy space previously inaccessible to fixed-regularization methods.

## A.1 Limitations

While ADRPO demonstrates significant improvements over existing approaches, several limitations should be acknowledged. Our experiments, though comprehensive, were primarily conducted on models of moderate scale (SD3 2B, Qwen2-0.5B, and Qwen3-0.6B) due to computational constraints. An important avenue for future work is extending these findings to much larger foundation models, where the exploration-exploitation dilemma becomes even more critical. Particularly, models with parameters in the hundreds of billions are known to be more susceptible to training collapse during online RL fine-tuning, potentially making adaptive regularization even more crucial at scale.

The advantage-based formulation introduces a slight computational overhead compared to purely reward-weighted methods like ORW-CFM-W2 [14] and RAFT [9]. Though we mitigate this through efficient batch-based normalization techniques similar to those used in GRPO [32], further optimization could reduce this overhead. Our current implementation of advantage estimation using batch statistics works well in practice but could be improved with more sophisticated value approximation methods, especially for complex reward landscapes with high variance.

## A.2 Broader Impact

In general, ADRPO represents a significant advance in reinforcement learning fine-tuning methodology with potential impacts extending beyond the specific models tested. The ability to efficiently navigate the exploration-exploitation trade-off through adaptive regularization addresses a fundamental challenge in the field, potentially influencing how the broader research community approaches model alignment.

Our finding that a relatively small model (2B parameters) can outperform substantially larger models (4.8B and 12B parameters) when fine-tuned with ADRPO has important implications for personal access to high-quality generative AI. This parameter efficiency could significantly reduce the computational resources required for state-of-the-art performance, making advanced generative capabilities more accessible to researchers and organizations with limited resources while reducing the environmental footprint of both training and deploying such systems.

The emergent ability to escape local optima demonstrated in our LLM experiments suggests that advantage-guided adaptive regularization may unlock previously unattainable regions of policy space. This capability could inspire new approaches to optimization in high-dimensional spaces where fixed regularization schemes tend to converge prematurely to a sub-optimal.

Our work also introduces a unified framework that bridges continuous and discrete generative paradigms, offering a consistent solution to the exploration-exploitation dilemma across domains. This cross-domain applicability could foster greater knowledge transfer between previously disparate research communities working on different types of generative models.

# B Experimental Details

In this section, we provide comprehensive details of our experimental setup for both text-to-image alignment and language model fine-tuning tasks. To maintain clarity, we present these domains separately.

## B.1 Flow Matching Model Fine-tuning Tasks

### B.1.1 Baseline Methods

**Base Model**  We adopt Stable Diffusion 3 (SD3) [13] as our base model—a 2B parameter architecture that combines a Multimodal Diffusion Transformer (MMDiT) backbone with a rectified flow training objective. SD3 represents the latest evolution of latent diffusion models, introducing a joint text-image Transformer that enables rich bidirectional attention between prompt tokens and image latents. Unlike earlier score-based approaches, SD3 is trained via conditional flow matching under a rectified trajectory, where the model learns to predict direct velocity fields between noise and data, improving sample efficiency and semantic alignment. It leverages multiple frozen text encoders (CLIP and T5) and improved autoencoding for high-resolution image synthesis, while achieving state-of-the-art performance on prompt fidelity, compositional reasoning, and text rendering. This combination of high quality, controllability, and architectural flexibility makes SD3 a robust and representative base model for studying the effects of reinforcement learning fine-tuning, such as ADRPO.

**Larger-Scale Flow Matching Models**  To comprehensively evaluate ADRPO against parameter scaling approaches, we selected two state-of-the-art text-to-image diffusion models that leverage flow-matching training objectives and significantly larger parameter counts than our base model. These models serve as strong upper baselines in terms of both capacity and generation quality:

1. **FLUX.1-Dev** [43], with 12B parameters, represents the high-performance frontier of open-source flow-matching architectures. It employs a rectified flow training objective based on Wasserstein-2 optimal transport, enabling more stable and efficient training compared to traditional score-matching methods. FLUX integrates a multimodal diffusion transformer (MMDiT) with powerful prompt conditioning mechanisms, achieving near-photorealistic output, superior compositional fidelity, and high stylistic diversity. It is widely regarded as one of the most capable open models in terms of prompt adherence, fine-grained semantic alignment, and artistic control.

2. **SANA-1.5** [38], with 4.8B parameters, serves as a strong intermediate-scale baseline. It introduces an efficient diffusion transformer architecture that combines linear attention mechanisms and a highly compressed autoencoder (32x downsampling), enabling high-resolution generation at lower computational cost. SANA adopts a decoder-style language model for text conditioning and achieves state-of-the-art results on the GenEval benchmark for prompt-image alignment. Despite its moderate size, SANA.1.5 offers a competitive trade-off between generation quality, efficiency, and controllability.

Both models exemplify the benefits of scaling parameter count and architectural sophistication within the flow-matching paradigm. By comparing against them, we isolate the advantages of ADRPO in improving alignment and control without relying solely on larger models. This allows us to highlight ADRPO's efficiency and generalization capabilities, especially when applied to smaller models such as our 2B-parameter SD3 baseline.

**RL Baseline Methods**  For fine-tuning method comparisons, we included approaches representing the most representative spectrum of current techniques:

**RAFT** [9] implements a reward-ranked fine-tuning approach that selects high-quality outputs based on reward scores, providing a online RL baseline. This approach has demonstrated considerable effectiveness in improving generative models but lacks the adaptive divergence regularization mechanisms essential for preserving model capabilities during policy optimization.

**DPO** [37] adapts the Direct Preference Optimization method to flow matching models, providing an established offline RL baseline. We apply diffusion-DPO to flow matching following methodologies

established in recent literature [8]. This approach offers stable optimization and effective diversity preservation through its implicit regularization properties, though it may be limited in its ability to explore the full policy space due to its offline nature. Given DPO's widespread adoption and demonstrated success across various fine-tuning tasks, it serves as our primary offline RL fine-tuning baseline.

**ORW-CFM-W2** [14] represents the current state-of-the-art in online RL fine-tuning for flow matching models, employing fixed Wasserstein-2 regularization combined with reward weighting. As the first online RL fine-tuning method developed specifically for flow matching models, it achieves leading performance in this domain through its W2 regularized online RL framework. This method provides a crucial benchmark against which to evaluate our ADRPO approach, as it represents the online SOTA method with fixed Wasserstein-2 regularization, allowing us to directly highlight the effectiveness of our proposed adaptive divergence regularization mechanism.

### B.1.2    Reward Models and Evaluation

For text-to-image generation, we implemented a comprehensive reward system combining multiple complementary models to evaluate different aspects of generation quality:

1. **Reward Model.** We used CLIP Score [25] to compute cosine similarity between text prompts and generated images (ClipScore) as our reward model for all text-to-image alignment task.

2. **Quality Assessment.** We employed a aesthetic predictor to evaluate visual appeal from ImageReward [39].

3. **Human Preference Models.** We incorporated ImageReward [39], trained on direct human judgments, and PickScore [21], developed through large-scale pick-one-from-four preference data, to align our generations with human aesthetic preferences. We complemented this with BLIP-based [12] evaluation to mitigate architecture-specific biases.

4. **Diversity Evaluation.** For diversity evaluation, we developed ClipDiversity, which measures the average pairwise distance between CLIP embeddings of multiple generated images of current FM model.

### B.1.3    Prompt Datasets

Our text-to-image experiments utilized a diverse collection of prompts selected to evaluate different capabilities. DrawBench [28] provided our primary test set, covering attribute binding, spatial relationships, counting accuracy, and text rendering. We extended this with artistic style prompts from RAFT [9] (e.g., "Van Gogh style astronaut") and custom compositional prompts testing multi-object relationships (e.g., "A green apple and a black backpack").

## B.2    LLM Fine-tuning Tasks

### B.2.1    Baseline Methods

**Base Models**    For our language model experiments, we employed Qwen2 [41] and Qwen3 [42] models as base architectures, representing recent advancements in autoregressive LLM models. These models demonstrate strong foundational capabilities and serve as robust pre-trained reference model for fine-tuning experiments. Specifically, Qwen2 and Qwen3 incorporate key architectural enhancements such as Grouped Query Attention (GQA), RoPE positional encoding, and long-context support (up to 128K for Qwen3), which contribute to efficient inference and robust context handling. Moreover, despite their relatively small parameter sizes (0.5B and 0.6B respectively), these models exhibit competitive performance across a range of reasoning, code generation, and language understanding benchmarks. Their strong pretraining on diverse multilingual and domain-specific corpora—including high-quality instructional data, code, and math—ensures excellent generalization. Qwen3 further introduces a hybrid prompting paradigm that enables dynamic switching between direct answering and step-by-step reasoning, significantly enhancing the model's flexibility and interpretability during instruction-following tasks. These strengths make Qwen2 and Qwen3 especially well-suited for fine-tuning via reinforcement learning from human feedback (RLHF), where high-quality priors and reasoning ability are essential for aligning model behavior with human preferences.

**RL Baseline Methods** For RL fine-tuning comparison, we selected GRPO [32] with fixed KL regularization ($\beta = 0.04$ [32]) as it represents the current state-of-the-art in online RL fine-tuning for LLMs. GRPO improves upon earlier methods like PPO [31] through group-level advantage estimation and more efficient policy optimization. However, it crucially still relies on static regularization that treats all samples equally regardless of their quality or uncertainty. This limitation makes GRPO an ideal candidate for demonstrating the advantages of our adaptive regularization approach, as both methods share the same underlying optimization framework but differ specifically in their treatment of regularization (also can be served as our ablation studies).

### B.2.2 Reward Model and Evaluation

For LLM fine-tuning, we used RM-Gemma-2B [27, 9], a reward model built upon the Gemma-2B language model and fine-tuned using a diverse collection of human preference datasets. RM-Gemma-2B maps input completions to scalar reward values, which serve as proxy signals for alignment with human preferences. The model is trained using pairwise comparison data spanning a wide range of tasks—including helpfulness, harmlessness, factuality, and reasoning—through a Bradley-Terry style objective that encourages higher scores for preferred responses. This formulation enables the reward model to capture nuanced quality differences across candidate outputs. To support more stable and informed policy updates, we further incorporated entropy-based regularization to evaluate and balance the exploration-exploitation dynamics of the fine-tuned policies. This combined approach ensures that the optimization process not only aligns outputs with human values but also maintains diversity and adaptability in model behavior.

### B.2.3 Prompt Datasets

For the large language model fine-tuning, we have used the RLHFlow/test_generation_2k dataset [10], containing 2,000 diverse prompts compiled from high-quality instruction-following datasets, and we randomly choose 10% as test prompts. This diverse prompt set allowed comprehensive evaluation across multiple dimensions, including factual accuracy, reasoning capabilities, and response quality. Specifically, the prompts were drawn from a combination of several representative and complementary sources: **UltraFeedback** [6], **Capybara** [7], **UltraInteract** [45], and **OpenOrca** [22].

- **UltraFeedback** provides high-quality single-turn instruction-response pairs with rich feedback annotations generated by GPT-4, including multi-dimensional numerical scores (e.g., helpfulness, correctness, conciseness) and textual critiques. These annotations support fine-grained evaluation and reward modeling.

- **Capybara** contributes multi-turn dialogues generated through the Amplify-Instruct pipeline, which enriches single-turn seed prompts into deep, logically consistent conversations. It emphasizes diverse topics, natural phrasing, and contextual reasoning, making it valuable for evaluating sustained dialogue coherence.

- **UltraInteract** focuses on complex tasks involving step-by-step reasoning, such as math, coding, and logic problems. Each example includes multi-step trajectories with intermediate model outputs, environment feedback, and correctness signals, enabling assessment of models' planning and iterative refinement abilities.

- **OpenOrca** offers a large-scale collection of instruction-response pairs distilled from GPT-4 and GPT-3.5 using the FLAN dataset collection. Its responses often include chain-of-thought style rationales, making it a useful benchmark for evaluating models' reasoning depth and informativeness.

By combining prompts from these datasets, the test set enables comprehensive evaluation of a model's capabilities across a wide range of real-world tasks and dialogue scenarios, from single-turn factual queries to multi-turn, multi-step reasoning challenges.

### B.3 Multi-Modal Audio Reasoning Tasks

### B.3.1 Baseline Methods

**Base Model** For our multi-modal reasoning experiments, we employed Qwen2.5-Omni-7B [40] (bf16 precision), a state-of-the-art multi-modal model capable of processing both audio and text inputs.

This model represents a significant advancement in audio-language understanding, incorporating a dual-tower architecture that separately encodes audio and text modalities before fusing them through cross-attention mechanisms. Despite its 7B parameter size, Qwen2.5-Omni demonstrates competitive performance on various audio understanding benchmarks through pre-training on diverse audio-text paired data including speech, music, and environmental sounds. The model's architecture enables fine-grained temporal reasoning and acoustic pattern recognition, making it well-suited for complex audio-visual question answering tasks requiring multi-step reasoning and contextual understanding.

**Baseline Methods**   For RL fine-tuning comparison, we selected GRPO [32] with fixed KL regularization ($\beta = 0.04$) as our primary baseline, consistent with our LLM experiments. GRPO's group-level advantage estimation and efficient policy optimization make it a strong baseline for multi-modal reasoning tasks. Additionally, we compared against commercial systems including Gemini 2.5 Pro [5] and GPT-4o Audio [19] to evaluate our method's competitiveness against substantially larger proprietary models. This comparison demonstrates whether adaptive regularization can enable smaller open-source models to match or exceed the capabilities of large-scale commercial systems.

### B.3.2   Reward Model and Evaluation

For multi-modal audio reasoning fine-tuning, we employed a dual reward mechanism combining accuracy rewards and format rewards, following the approach in DeepSeek-R1 [16]. Accuracy rewards are based on answer correctness, computed by exact matching between the model's predicted answer and the ground-truth answer from the dataset. Format rewards ensure that responses follow the expected structure with explicit thinking traces before final answers, encouraging the model to generate effective reasoning processes. This combined reward structure promotes both correctness and interpretable reasoning patterns.

We evaluated performance on the MMAU benchmark [29], a comprehensive multi-task audio understanding benchmark testing reasoning across three primary categories: sound understanding (environmental sounds, acoustic events), music understanding (genre, instruments, tempo), and speech understanding (speaker characteristics, linguistic content, prosody). The benchmark's 1k test-mini split provides diverse evaluation across these modalities, enabling assessment of generalization and reasoning capabilities.

### B.3.3   Dataset

For multi-modal audio reasoning fine-tuning, we used the AVQA dataset [44], which provides audio-visual question answering tasks requiring temporal reasoning and contextual understanding. The dataset contains diverse question types spanning spatial localization, temporal ordering, counting, and causal reasoning about audio-visual events. Each example consists of an audio clip (we extract them from the video), a natural language question, and multiple-choice answers, requiring models to integrate acoustic information with semantic understanding to arrive at correct conclusions. The training split contains various difficulty levels, from simple sound source identification to complex multi-step reasoning about temporal sequences and causal relationships. This diversity makes AVQA particularly suitable for evaluating whether adaptive regularization can enhance multi-modal reasoning capabilities compared to fixed regularization approaches. By fine-tuning on AVQA and evaluating on the independent MMAU benchmark, we assess both in-domain reasoning improvement and cross-domain generalization to unseen audio understanding tasks.

### B.4   Computation Resources

All experiments were conducted on NVIDIA A6000 (48GB) GPUs. For SD3 fine-tuning tasks [13], we employed parameter-efficient LoRA [18] adaptation to reduce memory requirements and training time, while still achieving excellent results. For the relatively smaller Qwen2-0.5B [41] and Qwen3-0.6B [42] language models, we performed direct full-parameter fine-tuning without LoRA. For Qwen2.5-Omni-7B [40], we utilized at least 10 A6000 GPUs to accommodate the model's multi-modal architecture and larger parameter size.

Our experimental setup utilized publicly available open-source reward/evaluation models and datasets across all domains, ensuring reproducibility and alignment with established benchmarks. The computation requirements varied significantly between tasks: LLM fine-tuning experiments (Qwen2/3) were

relatively efficient, typically completing within 12-24 hours per model configuration; SD3 fine-tuning tasks were more computationally intensive, requiring approximately 2-3 days; and multi-modal audio reasoning fine-tuning (Qwen2.5-Omni-7B) required 3-4 days due to the larger model scale and multi-modal processing complexity.

## C Algorithm Pseudocode

We first detail our algorithm pseudocode in Algorithm 1 for fine-tuning flow matching models (we use linear interpolation path as an example). Noting that, we can sample from current learned velocity field $\mathbf{v}_\theta(x_t, t, c)$ via solving: $x_1 = x_0 + \int_0^1 \mathbf{v}_\theta(x_t, t, c)dt$, wherein $x_0 \sim p(x_0)$ and $p(x_0)$ is a standard gaussian distribution [23, 13]. As for our method for fine-tuning LLM models, we can simply add an extra advantage-weighted KL divergence into the original GRPO training loss as Equ. (7), therefore it is easy to be implemented.

---

**Algorithm 1** Adaptive Divergence Regularized Policy Optimization (ADRPO) for SD3 Fine-tuning

---

**Require:** Pre-trained flow matching model $\pi_{\text{ref}}$ (SD3), baseline regularization coefficient $\beta_0$, advantage clipping range $[A_{\min}, A_{\max}]$, learning rate $\eta$

1: Initialize fine-tuned policy $\pi_\theta^0$ with pre-trained parameters (or LoRA adaptation)
2: **for** training iteration $n = 1, 2, \ldots$ **do**
3:      Sample a batch of text prompts $\{c_i\}_{i=1}^B \sim p(c)$
4:      Sample target states $\{x_1^i\}_{i=1}^B \sim \pi_\theta^{n-1}(x|c_i)$ from current policy ▷ Online sampling strategy
5:      **for** each prompt $c_i$ and its generated image $x_1^i$ **do**
6:          Compute reward $R(x_1^i, c_i)$ using CLIP Score
7:          Sample intermediate time step $t_i \sim \mathcal{U}(0, 1)$
8:          Compute intermediate state $x_t^i = (1 - t_i)x_0^i + t_i x_1^i$      ▷ Straight-line interpolation
9:          Compute target velocity $u_t^i = x_1^i - x_0^i$
10:     **end for**
11:     Compute baseline value $V(c_i) = \frac{1}{B} \sum_{i=1}^B R(x_1^i, c_i)$ for each context
12:     Compute advantage $A(x_1^i, c_i) = R(x_1^i, c_i) - V(c_i)$
13:     Apply advantage clipping: $A_{\text{clipped}}(x_1^i, c_i) = \text{clip}(A(x_1^i, c_i), A_{\min}, A_{\max})$
14:     Compute adaptive regularization coefficient $\beta_{\text{tot}} = \beta_0 - A_{\text{clipped}}(x_1^i, c_i)$
15:     Update model parameters using the ADRPO loss $\mathcal{L}_{\text{ADRPO-FM}}(\theta)$ from Equation (6):
16:        $\theta \leftarrow \theta - \eta \nabla_\theta \mathcal{L}_{\text{ADRPO-FM}}(\theta)$
17: **end for**
18: **return** Fine-tuned policy $\pi_\theta$

---

# D  Additional Experimental Results

## D.1  Flow Matching Model Fine-tuning Tasks

### D.1.1  Additional Qualitative Results

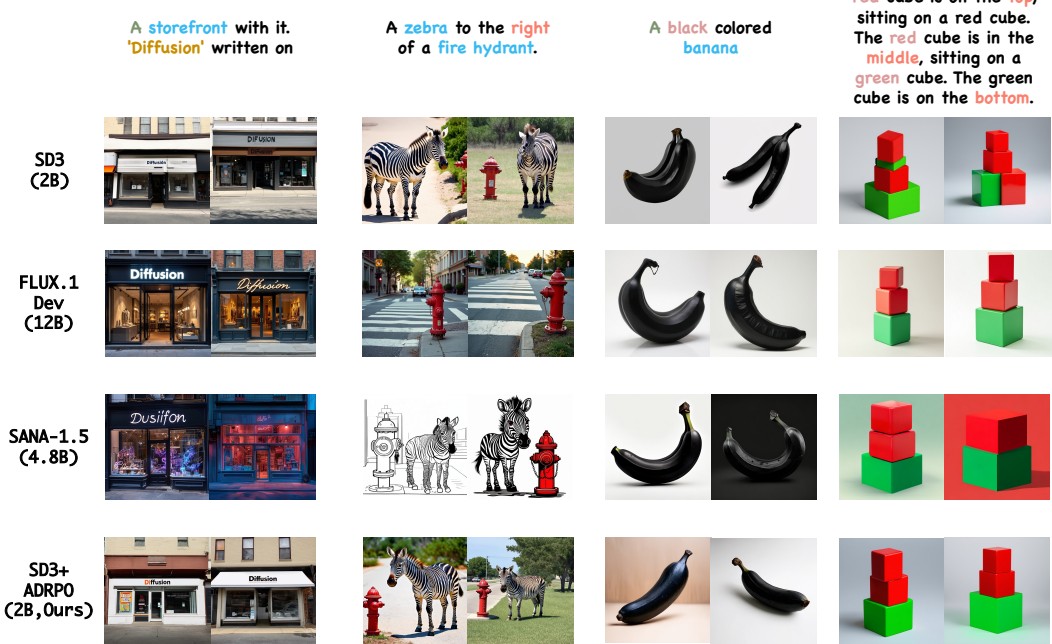

Figure 5: **Additional Qualitative Comparison with Large FM Generative Models.** Our ADRPO demonstrates superior performance in Attribute Binding, Coloring, Counting, Text Rendering and Position.

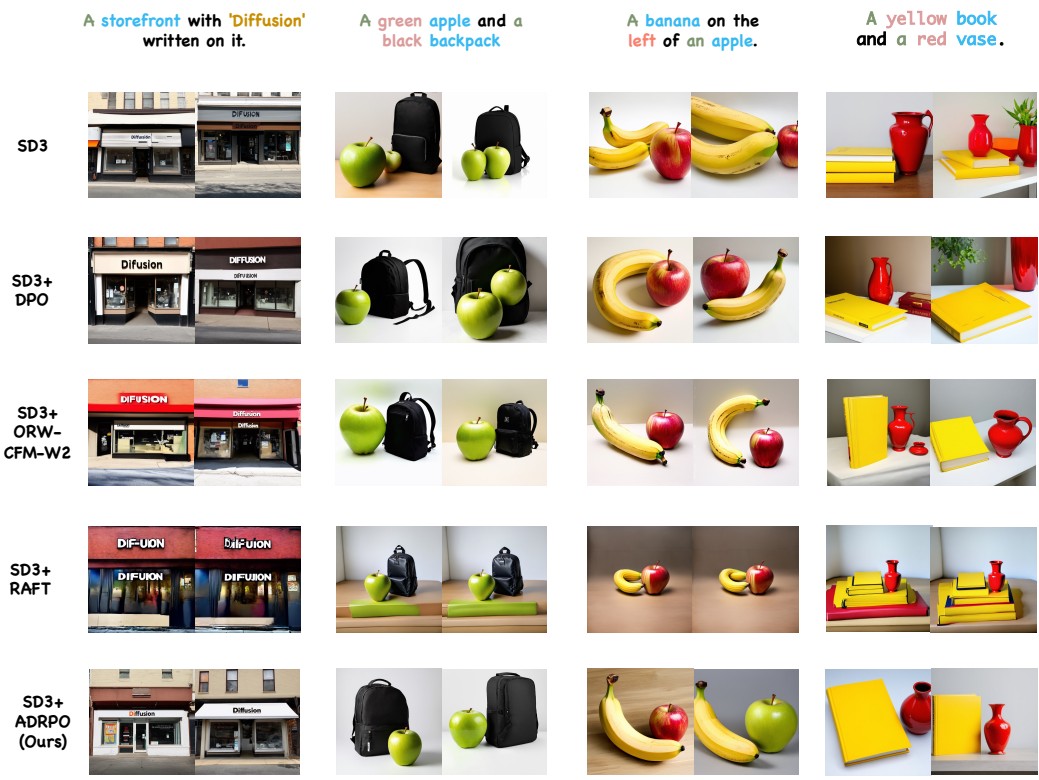

Figure 6: **Additional Qualitative Comparison with Other RL Fine-tuning Methods.** Our ADRPO demonstrates superior performance in Text Rendering, Attribute Binding, Coloring, Counting and Position.

### D.1.2 Learning Curves

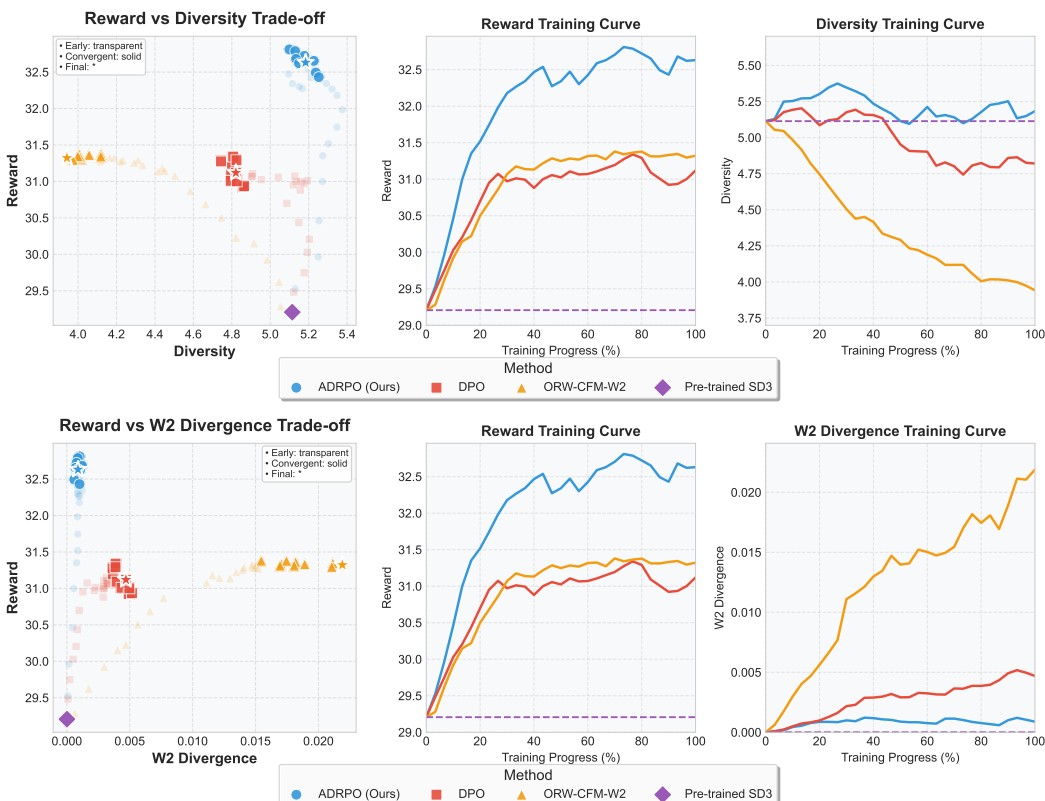

Figure 7: Learning Curves of Fine-tuning SD3. Left: Complete policy optimization trajectories across three different methods throughout training using a same seed (for fairness). Transparency indicates progression from early (transparent) stages through convergent (solid) to final (star) checkpoints, with each point representing a learned policy from different iterations. Center and right: Learning curves of RL agents.

## D.2 LLM Fine-tuning Tasks

### D.2.1 Learning Curves

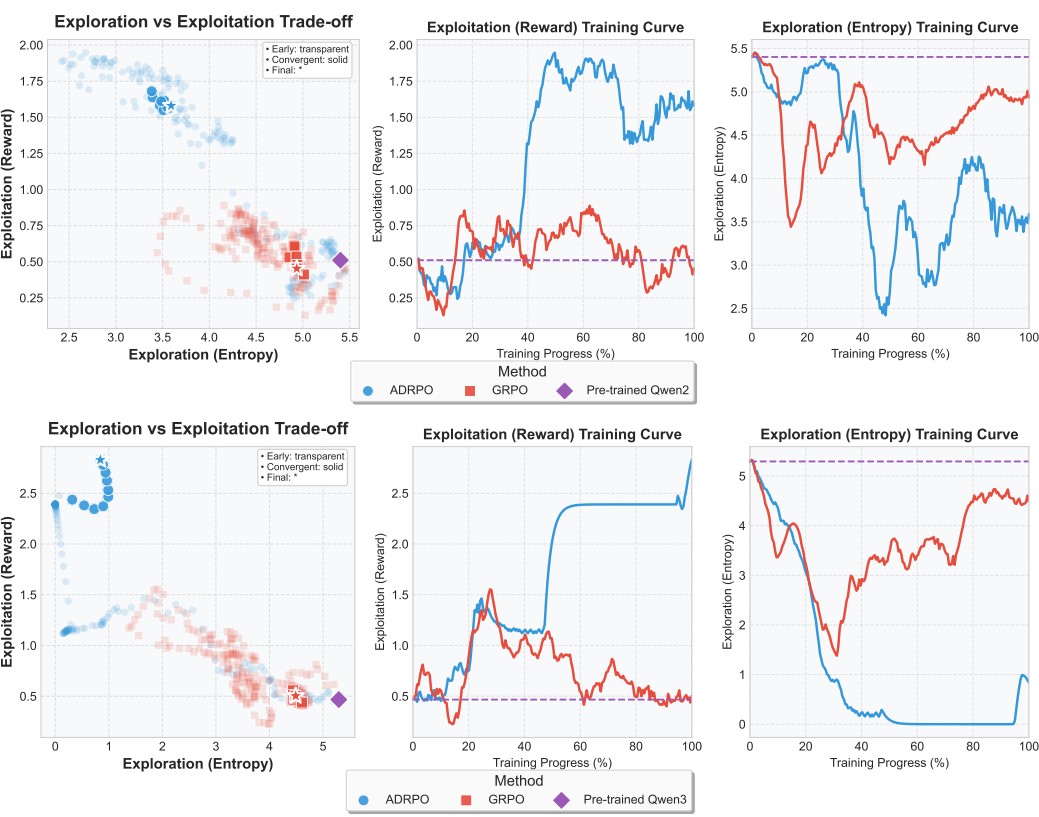

Figure 8: Learning Curves of LLM Fine-tuning Experiments (100 iterations, no early stop).

## D.2.2 Reward and KL Divergence Trade-off

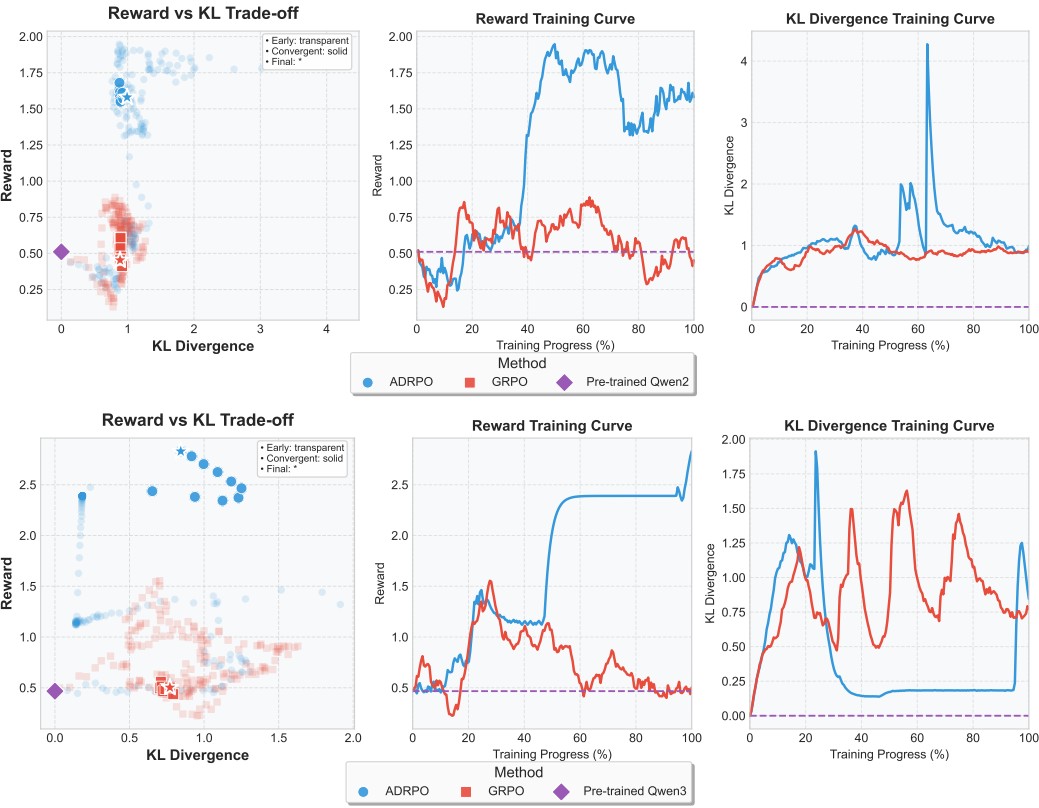

Figure 9: Reward Divergence Trade-off of LLM Fine-tuning Experiments (100 iterations, no early stop).

### D.3 Multi-Modal Audio Reasoning Tasks

To further demonstrate ADRPO's versatility beyond text-to-image generation and text-only LLMs, we conducted additional experiments fine-tuning the multi-modal reasoning model Qwen2.5-Omni-7B [40] on audio understanding tasks. This evaluation showcases ADRPO's effectiveness in a completely different domain requiring complex multi-modal reasoning capabilities.

#### D.3.1 Experimental Setup

We fine-tuned Qwen2.5-Omni-7B (bf16 precision) on the AVQA dataset [44], which provides audio-visual question answering tasks requiring temporal reasoning and contextual understanding. The model was optimized using verifiable rewards based on answer correctness and format rewards for proper response structure. We evaluated performance on the MMAU benchmark (1k test-mini split) [29], a comprehensive multi-task audio understanding benchmark that tests reasoning across sound, music, and speech understanding.

#### D.3.2 Quantitative Results

Tab. 4 presents our results on the MMAU benchmark (1k test-mini split), comparing ADRPO against the base model, baseline RL method (GRPO), and strong commercial models. Our method achieves substantial improvements across all audio reasoning categories, with particularly strong gains in sound understanding (+9.61% over base model) and speech understanding (+6.91% over base model).

Table 4: Performance comparison on MMAU benchmark for audio reasoning tasks (1k test-mini split) [29]. ADRPO significantly outperforms both baseline RL methods and larger commercial models across all categories.

| Method | Sound (%) | Music (%) | Speech (%) | Total Accuracy (%) |
|---|---|---|---|---|
| Qwen2.5-Omni-7B (base) | 72.37 | 64.37 | 69.07 | 68.6 |
| GRPO (baseline RL) | 77.18 | 70.66 | 74.77 | 74.2 |
| Gemini 2.5 Pro | 75.08 | 68.26 | 71.47 | 71.6 |
| GPT-4o Audio | 64.56 | 56.29 | 66.67 | 62.5 |
| **ADRPO (Ours)** | **81.98** | **70.06** | **75.98** | **76.0** |

Notably, our ADRPO fine-tuned 7B model outperforms substantially larger commercial models including Gemini 2.5 Pro and GPT-4o Audio, demonstrating the effectiveness of adaptive regularization across different model scales and architectures. The consistent improvements over GRPO (+1.8% total accuracy) validate that our advantage-based adaptive mechanism provides benefits beyond text generation domains.

#### D.3.3 Qualitative Analysis

Beyond quantitative metrics, we analyzed the reasoning processes of ADRPO and GRPO to understand how adaptive regularization improves multi-modal understanding. Tab. 5 presents representative examples where ADRPO demonstrates superior step-by-step reasoning compared to fixed regularization methods.

These examples reveal that ADRPO enables more accurate temporal sequence understanding, better source identification through attention to acoustic characteristics, and superior contextual reasoning that avoids conflating multiple environmental cues. The fixed regularization in GRPO leads to premature convergence on incorrect answers, while ADRPO's adaptive mechanism allows the model to better explore the reasoning space and arrive at correct conclusions.

#### D.3.4 Ablation on Advantage Clipping

To understand the robustness of ADRPO to hyperparameter choices in this new domain, we conducted ablation studies on different advantage clipping ranges. Tab. 6 shows results with different clipping configurations.

The results demonstrate that ADRPO achieves consistent improvements over GRPO across all clipping configurations, with performance variations of less than 0.4% between different settings.

Table 5: Qualitative comparison of reasoning traces between ADRPO and GRPO on challenging audio reasoning tasks. ADRPO demonstrates more accurate temporal analysis and contextual understanding.

| Question | ADRPO Response | GRPO Response |
|---|---|---|
| **Temporal Sequence:** What was the last sound in the sequence? *Choices: footsteps, dog_barking, camera_shutter_clicking, tapping_on_glass* | The sequence of sounds includes footsteps, a dog barking, a camera shutter clicking, and a tapping on glass. The last sound is a distinct 'tapping on glass', which is a clear and unique sound compared to the others. **Answer: tapping_on_glass** ✓ | The last sound in the sequence is a camera shutter click, which is a distinct sound from the other options provided. **Answer: camera_shutter_clicking** × |
| **Source Identification:** Based on the given audio, identify the source of the flowing water. *Choices: Bathtub, River, Fountain, Rain* | The audio suggests a continuous, steady flow of water, which is characteristic of a bathtub. The other options (river, fountain, rain) typically have more varied or intermittent sounds. **Answer: Bathtub** ✓ | The sound is consistent with a steady flow of water, likely from a man-made source like a bathtub or fountain rather than natural sources like a river or rain. **Answer: Fountain** × |
| **Contextual Reasoning:** Considering the information in the audio, what type of weather condition can be inferred? *Choices: Windy, Calm, Rainy, Snowy* | The presence of wind noise suggests a windy condition. The absence of rain or snow sounds indicates it's not rainy or snowy. **Answer: Windy** ✓ | The presence of wind noise and the sound of rain hitting a surface suggest a windy and rainy weather condition. **Answer: Rainy** × |

Table 6: Ablation study on advantage clipping ranges for audio reasoning tasks. All ADRPO configurations substantially outperform GRPO baseline, with stable performance across different clipping settings.

| Clipping Range | $A_{\min}$ | $A_{\max}$ | Sound (%) | Music (%) | Speech (%) | Total (%) |
|---|---|---|---|---|---|---|
| $1 \times \beta_0$ (recommended) | -0.04 | 0.04 | 81.98 | 70.06 | 75.98 | **76.0** |
| $2 \times \beta_0$ | -0.08 | 0.08 | 84.08 | 69.46 | 73.57 | 75.7 |
| $0.5 \times \beta_0$ | -0.02 | 0.02 | 82.58 | 71.26 | 74.47 | 76.1 |
| GRPO (baseline) | - | - | 77.18 | 70.66 | 74.77 | 74.2 |

This stability confirms that the fundamental advantage lies in having adaptive regularization versus fixed regularization, rather than in precise hyperparameter tuning. The $2 \times \beta_0$ setting achieves the highest sound understanding accuracy (84.08%) but shows reduced performance on music and speech tasks, suggesting that excessive negative regularization can lead to overly aggressive exploitation that hurts generalization. Our recommended $1 \times \beta_0$ setting provides the best balance across all modalities.

These multi-modal audio reasoning experiments provide several key insights: (1) ADRPO's effectiveness extends beyond visual and text generation to complex multi-modal reasoning tasks; (2) the advantage-based adaptive mechanism enables superior step-by-step reasoning and reduces failure modes compared to fixed regularization; (3) ADRPO demonstrates robust performance across different hyperparameter settings, making it practical for new domains; and (4) our method enables smaller models (7B) to outperform much larger commercial models through more effective exploration-exploitation balance.

