# OpenReview forum: "Adaptive Divergence Regularized Policy Optimization for Fine-tuning Generative Models"
_NeurIPS.cc/2025/Conference — NeurIPS 2025 poster_

### Official Review · Reviewer_oj6J · 2025-06-21

**Clarity:** 3
**Significance:** 3
**Originality:** 3
**Rating:** 5
**Confidence:** 4

**Summary:**

This paper proposes an adaptive divergence regularization technique that effectively improves reward fine-tuning results by introducing per-sample divergence control, which provides more flexibility to the optimization process. Essentially, it strengthens divergence regularization for samples of lower quality (in terms of the target reward) and relaxes for those of higher quality. This method works well for both text-to-image generation (continuous domain FM velocity-based divergence regularization) and LLM (classic KL divergence on discrete domain).

**Questions:**

+ Wouldn't the reward hacking be worse if the proposed approach relaxes divergence control for samples that are highly aligned with the underlying reward? For instance, say the target reward model gives a high reward to certain visual artifacts, and stronger divergence control helps to mitigate this.

**Ethical Concerns:**

["NO or VERY MINOR ethics concerns only"]

**Final Justification:**

I keep my original score.

**Limitations:**

See "weakness" and "questions"

**Quality:**

3

**Strengths And Weaknesses:**

Strengths:
+ Good motivation for an important problem in reward fine-tuning generation models (the exploitation vs exploration dilemma).
+ Results look convincing (especially Fig. 3 & 4).

Weakness:
+ I would like to see more visual results, including the failure cases of the proposed method, to understand the exploitation vs exploration dilemma better and how the method improves the generation quality on the hard cases.
+ Besides ClipDiversity and W2 divergence as the y-axis (say the x-axis shows reward scores), can the authors also show the plot with FIDs (in the style of Fig. 8 from the appendix of [1])? It better reflects image fidelity. As discussed in [1], the exploitation vs exploration dilemma in fine-tuning image generation models often comes down to reward scores vs. image fidelity. Maybe the authors can add a discussion on this matter.

[1] Reward Fine-Tuning Two-Step Diffusion Models via Learning Differentiable Latent-Space Surrogate Reward, CVPR 2025

---

> ### Author Rebuttal · Authors · 2025-07-30
>
> We are extremely grateful for your valuable and detailed review. Your feedback is instrumental in refining our work's impact and presentation. We especially appreciate your recognition of our method's strong motivation for addressing the exploitation-exploration dilemma in reward fine-tuning and your assessment of our convincing results. The suggestions for additional visual analyses, FID, and discussions on reward hacking provide valuable directions for strengthening our impact and clarity. We address your insightful questions below and will incorporate these clarifications and analysis into our final revision.
>
> > **Q1: I would like to see more visual results, including the failure cases of the proposed method, to understand the exploitation vs exploration dilemma better and how the method improves the generation quality on the hard cases.**
>
> **A1**: Thank you for this valuable feedback. We believe that ADRPO's improvements in generation quality and alignment can be best demonstrated by comparing our method to the failure modes of exploration-heavy/exploitation-heavy methods. We have provided extensive qualitative analysis in Figures 2 and 6, which clearly demonstrate these trade-offs and failure modes compared to our ADRPO.
>
> **Failures of Exploration-Heavy Methods (Poor Semantic Alignment)**: DPO maintains good generation diversity but consistently fails at semantic alignment. In Figure 2, DPO fails to render text properly ("Diffusion" appears incorrectly), generates anatomically incorrect zebras with malformed legs, and produces black bananas that lack realistic texture. In Figure 6, DPO demonstrates severe compositional failures—generating multiple apples instead of "A green apple" and misplacing objects in spatial relationship tasks.
>
> **Failures of Exploitation-Heavy Methods (Diversity and Quality Collapse)**: ORW-CFM-W2 aggressively pursues reward optimization but suffers from significant quality degradation. In Figure 2, it generates zebras with incorrect anatomy (extra limbs), fails at text rendering with blurred "Diffusion" text, and produces unnatural black bananas with poor texture quality. In Figure 6, exploitation-focused method RAFT exhibits similar failures with poor text rendering, overexposed images, and distorted bananas, struggling with compositional accuracy and visual naturalness across multiple prompts.
>
> **ADRPO's Superior Balance**: Our method successfully resolves these fundamental trade-offs by achieving both excellent semantic alignment and generation diversity simultaneously. In Figure 2, ADRPO correctly renders "Diffusion" text, generates anatomically accurate zebras, and produces realistic black bananas with proper texture. Figure 6 demonstrates ADRPO's superior performance on compositional challenges—correctly generating single objects, maintaining proper spatial relationships, and achieving accurate attribute binding while preserving visual quality.
>
> This comprehensive comparison demonstrates that ADRPO's adaptive regularization mechanism enables optimal exploration-exploitation balance, avoiding both semantic alignment failures of exploration-focused methods and the quality degradation issues that plague exploitation-focused methods.
>
> **Performance on Hard Reasoning Tasks**: Beyond visual generation, our additional experiments on fine-tuning Qwen2.5-Omni-7B [1] (bf16 precision) with AVQA dataset [2] demonstrate ADRPO's superior capabilities on complex reasoning scenarios that require multi-modal understanding and multi-step analysis. On the challenging MMAU benchmark [3], ADRPO achieves 76% total accuracy vs 74.2% for GRPO, with particularly strong improvements in complex audio reasoning (more will be included in our final revision).
>
> **Qualitative Analysis of Hard Reasoning Cases**:
>
> **Complex Temporal Analysis**:
>
> **Sample ID:** b7701ab1-c37e-49f2-8ad9-7177fe0465e9
> - **Question**: "What was the last sound in the sequence? Choices: footsteps, dog_barking, camera_shutter_clicking, tapping_on_glass"
> - **ADRPO**: "The sequence of sounds includes footsteps, a dog barking, a camera shutter clicking, and a tapping on glass. The last sound is a distinct 'tapping on glass', which is a clear and unique sound compared to the others." → **tapping_on_glass** ✓
> - **GRPO**: "The last sound in the sequence is a camera shutter click, which is a distinct sound from the other options provided." → **camera_shutter_clicking** ✗
>
> **Complex Contextual Reasoning**:
>
> **Sample ID:** 6d1ab354-944d-4155-a4ec-c851fbcb7c93
> - **Question**: "Considering the information in the audio, what type of weather condition can be inferred? Choices: Windy, Calm, Rainy, Snowy"
> - **ADRPO**: "The presence of wind noise suggests a windy condition. The absence of rain or snow sounds indicates it's not rainy or snowy." → **Windy** ✓
> - **GRPO**: "The presence of wind noise and the sound of rain hitting a surface suggest a windy and rainy weather condition." → **Rainy** ✗
>
> These challenging cases demonstrate that ADRPO enables superior step-by-step reasoning with more accurate temporal analysis, contextual understanding, and multi-modal integration, while GRPO consistently fails on complex reasoning tasks that require nuanced audio comprehension and logical inference.
>
> > **Q2: Besides ClipDiversity and W2 divergence as the y-axis (say the x-axis shows reward scores), can the authors also show the plot with FIDs (in the style of Fig. 8 from the appendix of [1])? It better reflects image fidelity. As discussed in [1], the exploitation vs exploration dilemma in fine-tuning image generation models often comes down to reward scores vs. image fidelity. Maybe the authors can add a discussion on this matter.**
>
> **A2**: We appreciate this suggestion about including FID metrics. However, it is worth noting that FID typically requires a ground-truth dataset for comparison, which is not available for the base models (SD3, SANA, etc.) and baselines we're comparing as their training data is usually different and hidden. During training, we only have prompts and train based on data generated by the model itself, rather than from ground-truth image datasets. Many other RL post-training works (such as ORW-CFM-W2 [4], DDPO [5] and DPOK [6]) also do not include FID metrics in their evaluations.
>
> **Alternative Quality Assessment**: While we cannot provide FID, the image quality metrics in Table 1 and visual results in Figures 2 and 6 address the fidelity concerns you raise. Our comprehensive evaluation includes multiple quality dimensions: Aesthetic scores (5.53→6.27), BLIPScore improvements (0.501→0.567), and human preference metrics (ImageReward: 0.97→1.61, PicScore: 20.81→22.78). These metrics, combined with our qualitative results, demonstrate that ADRPO achieves superior generation quality and semantic alignment compared to baseline methods and even much larger models (FLUX.1-Dev 12B, SANA-1.5 4.8B).
>
> **Reward, Quality and Diversity Balance**: Our training curves in Figure 3 demonstrate that ADRPO achieves better exploration-exploitation trade-offs by finding superior convergence points that maintain both high reward and diversity, suggesting improved fidelity-reward balance compared to methods that suffer from diversity collapse (ORW-CFM-W2) or reward plateau (DPO). The visual evidence in Figures 2 and 6 confirms this balance - our method avoids the overexposure artifacts and quality degradation seen in exploitation-heavy methods while maintaining more precise semantic alignment compared to exploration-heavy approaches.
>
> > **Q3: Wouldn't the reward hacking be worse if the proposed approach relaxes divergence control for samples that are highly aligned with the underlying reward? For instance, say the target reward model gives a high reward to certain visual artifacts, and stronger divergence control helps to mitigate this.**
>
> **A3**: Thank you for this insightful question. However, our adaptive mechanism actually provides better protection against reward hacking than fixed regularization methods through several key safeguards.
>
> **Advantage-Based Adaptive Regularization**: We use advantage estimates rather than raw rewards to adjust regularization strength. Our advantage estimation provides much more stable and normalized values compared to raw rewards, which can have high variance and extreme values. This advantage-based approach prevents drastic regularization changes that could occur with raw reward-based adjustment, maintaining more stable training dynamics.
>
> **Advantage Clipping Protection**: As detailed in Section 4.1, we employ advantage clipping $A_{clipped}= \text{clip}(A, A_{min}, A_{max})$ that **constrains regularization adjustments within reasonable bounds**. This prevents extreme regularization reduction even for samples with very high advantages while maintaining training stability.
>
> **Empirical Evidence Against Reward Hacking**: Figure 3 demonstrates that ADRPO achieves high reward performance **without the diversity collapse** characteristic of reward hacking seen in ORW-CFM-W2. Our visual results in Figures 2 and 6 consistently avoid the overexposure artifacts, unnatural distortions, and quality degradation that typically characterize reward hacking in exploitation-heavy methods. Instead, ADRPO maintains both high semantic alignment and natural visual quality, suggesting effective mitigation of reward hacking while achieving genuine improvements in generation quality.
>
> [1] Xu, Jin, et al. "Qwen2.5-omni technical report." 2025.
>
> [2] Yang, P., et al. "Avqa: A dataset for audio-visual question answering on videos." 2022.
>
> [3] Sakshi, S., et al. "Mmau: A massive multi-task audio understanding and reasoning benchmark." 2024.
>
> [4] Fan, J., et al. "Online reward-weighted fine-tuning of flow matching with wasserstein regularization." 2025.
>
> [5] Black, K., et al. "Training diffusion models with reinforcement learning." 2023.
>
> [6] Fan, Y., et al. "Dpok: Reinforcement learning for fine-tuning text-to-image diffusion models." 2023.

---

### Official Review · Reviewer_sGRU · 2025-06-29

**Clarity:** 3
**Significance:** 3
**Originality:** 3
**Rating:** 4
**Confidence:** 3

**Summary:**

The paper introduces Adaptive Divergence Regularized Policy Optimization (ADRPO), a plug-and-play framework that dynamically adjusts the weights of divergence regularization in RLHF of generative models. Instead of using a fixed KL or Wasserstein-2 penalty, ADRPO modulates the coefficient on a per-sample basis, reducing regularization for high-advantage samples to encourage exploitation, and strengthening it for low-advantage samples to preserve stability and diversity. The authors implement ADRPO with W-2 regularization for text-to-image flow-matching models (SD3) and with KL regularization for large language models (LLMs, via GRPO). ADRPO enables a 2B SD3 model to outperform larger 4.8 B and 12 B models across alignment, style transfer, and compositional control, and outperform other RL fine-tuning methods. In LLM fine-tuning (Qwen2/3), ADRPO achieves higher reward-entropy trade-offs than fixed-$\beta$ GRPO.

**Questions:**

see weaknesses. and
1. In my opinion, $L_{D}$ can be combined with $L_{PG} (L_{RL})$to form $A * min(ratio - D_{kl}, clip(ratio, 1+ \epsilon, 1-\epsilon) - D_{kl})$ or $A* (\lVert v_{\theta} - u \rVert^2 - \lVert v_\theta - v_{ref}\rVert^2)$, and plus a fixed KL penalty, could some adjustment to the ratio achieve a similar effect as described in the paper?
2. In fact, [1] mentions a similar idea of an Adaptive KL Penalty Coefficient. Discussing the differences between the approach in this paper and that in [1] would be beneficial for highlighting the originality of the current work.

[1] Schulman, John, et al. "Proximal policy optimization algorithms." arXiv preprint arXiv:1707.06347 (2017).

**Ethical Concerns:**

["NO or VERY MINOR ethics concerns only"]

**Final Justification:**

I thank the authors for addressing my concerns. I have no further questions, and recommend a borderline accept.

**Limitations:**

yes.

**Quality:**

3

**Strengths And Weaknesses:**

strengths
1. The proposed method appears simple yet effective, making it easy to follow.
2. ADRPO’s sample-wise adaptation of divergence strength directly addresses the core dilemma in RL fine-tuning, yielding more stable and effective policy updates than fixed-$\beta$ methods.
3. The paper presents convincing results of fine-tuning SD3 using ADRPO, outperforming larger models. Additionally, compared to other RL fine-tuning methods in SD3, ADRPO achieves better performance in both reward and divergence metrics.
4. ADRPO integrates seamlessly into existing RLHF/RL-from-reward frameworks for both continuous (flow matching) and discrete (LLMs) generative paradigms, without additional networks or architectural changes.

weaknesses
1. The paper claims that the method is applicable to both diffusion (flow matching) models and LLMs. However, the experimental section only presents results for text-to-image tasks, lacking quantitative or qualitative experiments involving LLMs.
2. ADRPO introduces several new hyperparameters ($\beta_0, A_{max}, A_{min}$). A ablation over these settings (and the effect of different clipping ranges) would help practitioners tune ADRPO in new domains.

---

> ### Author Rebuttal · Authors · 2025-07-30
>
> We deeply appreciate your thorough and constructive review. Your feedback provides crucial insights that will strengthen both the technical presentation and practical applicability of our work. We are particularly grateful for your recognition of ADRPO's simplicity and effectiveness, its seamless integration into existing frameworks, and the convincing experimental results demonstrating our method's ability to enable smaller models to outperform larger counterparts. Your detailed questions about technical connections and suggestions for additional ablations are extremely helpful. Below we systematically address each of your concerns and will incorporate these clarifications and analysis into our final revision.
>
>
> > **Q1: The paper claims that the method is applicable to both diffusion (flow matching) models and LLMs. However, the experimental section only presents results for text-to-image tasks, lacking quantitative or qualitative experiments involving LLMs.**
>
> **A1**: Thank you for the insightful feedback. We want to clarify that our paper includes comprehensive LLM fine-tuning experiments in Section 4.5 and Figure 4, demonstrating ADRPO applied to Qwen2 and Qwen3 models with detailed quantitative results including reward-entropy trade-offs and complete learning curves in Figure 8 showing ADRPO's superior exploration-exploitation dynamics.
>
> **Extra Multi-Modal Reasoning LLM Experimental Results**: To further validate our method's effectiveness across diverse LLM applications, we provide additional experimental results on fine-tuning Qwen2.5-Omni-7B [1] (bf16 precision) for multi-modal audio reasoning tasks using AVQA training data [2] with verifiable rewards and format rewards. On the MMAU benchmark [3] (a complex multi-modal reasoning benchmark), ADRPO demonstrates substantial improvements in audio reasoning tasks:
>
> | Method | Sound (%) | Music (%) | Speech (%) | Total Accuracy (%) |
> |--------|-----------|-----------|------------|-------------------|
> | Qwen2.5-Omni-7B (Base) | 72.37 | 64.37 | 69.07 | 68.6 |
> | **ADRPO (Ours)** | **81.98** | **70.06** | **75.98** | **76.0** |
> | GRPO (baseline RL) | 77.18 | 70.66 | 74.77 | 74.2 |
> | Gemini 2.5 Pro | 75.08 | 68.26 | 71.47 | 71.6 |
> | GPT-4o Audio | 64.56 | 56.29 | 66.67 | 62.5 |
>
> Notably, our ADRPO fine-tuned 7B model significantly outperforms much larger commercial models including Gemini 2.5 Pro and GPT-4o Audio, demonstrating the effectiveness of our adaptive regularization approach across different model scales and architectures.
>
> **Qualitative Analysis - Superior Reasoning Capabilities**: Beyond quantitative improvements, we provide detailed qualitative analysis comparing thinking traces between ADRPO and GRPO, revealing that our method enables superior step-by-step reasoning capabilities (more results will be included in our final revision):
>
>
> **Sample ID:** b7701ab1-c37e-49f2-8ad9-7177fe0465e9
> - **Question**: "What was the last sound in the sequence? Choices: footsteps, dog_barking, camera_shutter_clicking, tapping_on_glass"
> - **ADRPO**: "The sequence of sounds includes footsteps, a dog barking, a camera shutter clicking, and a tapping on glass. The last sound is a distinct 'tapping on glass', which is a clear and unique sound compared to the others." → **tapping_on_glass** ✓
> - **GRPO**: "The last sound in the sequence is a camera shutter click, which is a distinct sound from the other options provided." → **camera_shutter_clicking** ✗
>
>
> **Cross-Architecture Validation**: These results, combined with our original experiments on autoregressive LLMs (Qwen2/3) and flow matching models (SD3), demonstrate that ADRPO provides consistent improvements across **three distinct model families**: (1) **flow matching models** for continuous generation, (2) **autoregressive LLMs** for discrete sequence generation, and (3) **multi-modal reasoning models** for complex audio-text understanding. This comprehensive validation confirms the broad applicability and effectiveness of our adaptive regularization approach across diverse architectures, modalities, and reasoning tasks.
>
>
> > **Q2: ADRPO introduces several new hyperparameters $\left(\beta_0, A_{\text {max }}, A_{\text {min }}\right)$. A ablation over these settings (and the effect of different clipping ranges) would help practitioners tune ADRPO in new domains.**
>
> **A2**: We appreciate this feedback and understand practitioners' concerns about hyperparameter tuning in new domains. To directly address the concern, we conducted comprehensive ablation studies in a completely new domain - multi-modal audio reasoning - to demonstrate ADRPO's practical transferability and provide concrete tuning guidance.
>
> **Simple Guidelines for Practitioners**: ADRPO introduces three hyperparameters that can be set with minimal tuning effort:
> 1. **β₀ (baseline regularization)**: Set equal to the fixed regularization coefficient used in existing methods (e.g., β₀=0.04 for GRPO, β₀=1 for SD3 fine-tuning)
> 2. **A_min, A_max (advantage clipping)**: Use symmetric ranges around zero: A_min = -β₀, A_max = β₀
>
> This design makes ADRPO a plug-and-play enhancement requiring no additional hyperparameter search beyond existing methods.
>
> **New Domain Validation**: To validate these guidelines beyond our main experiments, we applied ADRPO to fine-tune the multi-modal reasoning model Qwen2.5-Omni-7B [1] on the AVQA dataset [2] using verifiable and format rewards, with evaluation on the MMAU benchmark [3]. We tested various clipping configurations:
>
> | Clipping Range | A_min | A_max | Sound (%) | Music (%) | Speech (%) | Total Accuracy (%) |
> |----------------|-------|-------|-----------|-----------|------------|-------------------|
> | 1×β₀ (recommended) | -0.04 | 0.04 | 81.98 | 70.06 | 75.98 | **76.0** |
> | 2×β₀ | -0.08 | 0.08 | 84.08 | 69.46 | 73.57 | 75.7 |
> | 0.5×β₀ | -0.02 | 0.02 | 82.58 | 71.26 | 74.47 | 76.1 |
> | GRPO (baseline) | - | - | 77.18 | 70.66 | 74.77 | 74.2 |
>
> **Key Findings for New Domain Adoption**: This validation in an entirely different domain confirms that practitioners can safely apply our recommended 1×β₀ setting when transferring ADRPO to unseen domains. All ADRPO configurations significantly outperform the GRPO baseline (74.2%), with remarkably stable performance variations of less than 0.4% across different clipping ranges. This demonstrates that the fundamental advantage lies in having adaptive regularization versus fixed regularization, rather than in precise hyperparameter tuning. The advantage-based mechanism automatically adjusts regularization strength based on sample quality, making ADRPO inherently robust across domains and reducing the hyperparameter tuning burden for practitioners.
>
> > **Q3: In my opinion, $L_D$ can be combined with $L_{PG}(L_{RL})$ to form $A * \min(\text{ratio} - D_{kl}, \text{clip}(\text{ratio}, 1+\epsilon, 1-\epsilon) - D_{kl})$ or $A * (||v_\theta - u||^2 - ||v_\theta - v_{\text{ref}}||^2)$, and plus a fixed KL penalty, could some adjustment to the ratio achieve a similar effect as described in the paper?**
>
> **A3**: Thank you for this insightful suggestion. The suggested formulations represent a creative way to absorb adaptive divergence terms into the RL terms, which is an interesting  perspective.
>
> While these formulations might potentially achieve similar adaptive effects, we found our current decoupled formulation $\mathcal{L}_{RL} + (\beta_0 - A) \cdot \mathcal{L}_D$ (Equation 4) offers several key advantages: **clear interpretability** where the regularization strength $(\beta_0 - A)$ directly shows how advantage-based adaptation works, and **plug-and-play integration** that enables direct enhancement of existing methods like GRPO by simply replacing fixed $\beta$ with $(\beta_0 - A)$ without modifying core policy gradient computations. Our design prioritizes simplicity and practical applicability, allowing practitioners to easily adopt it in their current methods.
>
> > **Q4: In fact, [1] mentions a similar idea of an Adaptive KL Penalty Coefficient. Discussing the differences between the approach in this paper and that in [1] would be beneficial for highlighting the originality of the current work.**
>
> **A4**: Thank you for raising this important point. While PPO [1] does include an "adaptive KL penalty" variant, our approaches are fundamentally different in both mechanism and objective. **PPO operates at a global population level with rule-based adjustment to achieve a fixed target KL divergence $d_{\text{targ}}$ (a fixed hyperparameter throughout training), while our ADRPO performs sample-level adaptation to address the exploration-exploitation trade-off**.
>
> PPO's adaptive mechanism adopts simple heuristic rules (if KL < $d_{\text{targ}}$/1.5, then β ← β/2; if KL > $d_{\text{targ}}$ × 1.5, then β ← β × 2) to achieve a fixed target KL value $d_{\text{targ}}$ and applies the same β uniformly to all samples within each training iteration, regardless of individual sample quality. Critically, PPO's authors found this adaptive KL penalty approach performed significantly worse than their clipped objective (0.74 vs 0.82 in their experiments) and explicitly stated "the KL penalty performed worse than the clipped surrogate objective."
>
> In contrast, our ADRPO uses $\mathcal{L}^{ADRPO}(\theta) = \mathcal{L}^{RL}(\theta) + (\beta_0 - A) \cdot \mathcal{L}^D(\theta)$ to automatically differentiate between high-quality and low-quality samples, applying individualized regularization based on advantage estimates. This enables dynamic balancing of aggressive optimization for promising samples and conservative updates for inferior samples, achieving good performance across multiple domains as demonstrated in our experiments.
>
> [1] Xu, Jin, et al. "Qwen2.5-omni technical report." 2025.
>
> [2] Yang, Pinci, et al. "Avqa: A dataset for audio-visual question answering on videos." ACM MM. 2022.
>
> [3] Sakshi, S., et al. "Mmau: A massive multi-task audio understanding and reasoning benchmark." 2024.

---

> > ### Comment · Reviewer_sGRU · 2025-08-05
> > **discussion**
> >
> > I thank the authors for addressing my concerns. I have no further questions.

---

> ### Author Response · Authors · 2025-08-05
> **Thank You for Your Confirmation**
>
> Thank you for reviewing our rebuttal and confirming that our responses have adequately addressed your concerns. We sincerely appreciate your valuable and constructive feedback, which has helped us improve the technical presentation and practical applicability of our paper.

---

### Official Review · Reviewer_4P4K · 2025-06-30

**Clarity:** 3
**Significance:** 3
**Originality:** 3
**Rating:** 4
**Confidence:** 2

**Summary:**

The paper suggest to use adaptive regularization scheme for policy optimization enabling to balance between exploration and exploitation. The scheme called ADRPO is based on the advantage estimation - reducing regularization for high-value samples while applying stronger regularization to poor samples. The method is applied for online fine-tuning of LLMs (to extend DeepSeekMath’s GRPO idea), text to image alignment etc. The authors also suggest an advantage-weighted flow matching objective strengthening high-quality generations and pushing the model
away from poor generations. Numerical experiments (LLM fine-tuning and text to image generation) support suggested method.

**Questions:**

- I would suggest to add more explanation of advantage function estimation in particular for the section 3.4. ADRPO for Fine-tuning LLMs. At least summarise main ideas from the paper [29].
- I would also suggest to add more experiments replacing Qwen by other comparable models.
- Is it possible to overcome the problem of choosing baseline parameter $\beta_0$?

**Ethical Concerns:**

["NO or VERY MINOR ethics concerns only"]

**Final Justification:**

I thank the authors for their answers. I keep my score and stay positive.

**Limitations:**

Yes

**Paper Formatting Concerns:**

-

**Quality:**

3

**Strengths And Weaknesses:**

Strengths
- The paper is well written in easy to understand. It illustrates how different components work together.
 - Interesting idea of adaptive regularisation based on the advantage estimation. ADRPO automatically increases regularization for low-advantage (less promising) samples and decreases it for high-advantage (promising) samples, providing fine-grained control over the policy updates.
- ADRPO generalizes to both flow matching models (using Wasserstein-2 divergence) and LLMs (using KL divergence)
 - The experimental results look promising.

Weaknesses

- Additional computational costs due to advantage estimation. The paper would benefit from a more quantitative analysis of this overhead and practical guidance on how it scales with model size.
- While the empirical results are strong, the paper would be improved by discussing theoretical properties of the proposed adaptation.

---

> ### Author Rebuttal · Authors · 2025-07-30
>
> We sincerely thank you for your comprehensive and insightful review. Your feedback significantly enhances our work's presentation and clarity. We are grateful for your recognition of our adaptive regularization concept and its clear exposition, as well as the recognition of ADRPO's generalizability across different model architectures and the promising experimental outcomes. Your suggestions for   computational overhead and advantage estimation discussion are particularly valuable. We address your specific questions and concerns below and will incorporate these clarifications and analysis into our final revision.
>
> > **Q1: Additional computational costs due to advantage estimation. The paper would benefit from a more quantitative analysis of this overhead and practical guidance on how it scales with model size.**
>
> **A1:** Thank you for this insightful feedback. Our advantage estimation approach follows the group relative advantage estimation method used in GRPO, which does not require introducing additional critic models or value networks. For methods already using advantage estimation (like LLM fine-tuning with GRPO), ADRPO does not introduce too much additional computational overheads (i.e., plug and play). For tasks requiring new advantage estimation (like SD3 fine-tuning), we do not introduce substantial computational burden because we avoid additional critic networks. The main computational bottlenecks in these tasks are data rollout and reward model inferences, which exist across RL baseline methods, so our approach does not add significant extra computational cost. It is worth noting that group relative  advantage estimation has been widely used for being computation-efficient [1-2] and having excellent scalability properties [2].
>
> > **Q2: While the empirical results are strong, the paper would be improved by discussing theoretical properties of the proposed adaptation.**
>
> **A2:** Thank you for this insightful comments. The primary goal of our work is to propose a practically effective solution to improve existing RL fine-tuning methods by addressing key issues including exploration-exploitation trade-offs, reward hacking, and model collapse. Our extensive experiments demonstrate that the introduction of advantage-based dynamic regularization is indeed beneficial across diverse domains including LLMs (Figure 4), flow matching generative models (Table 1, Figures 1-3), and multi-modal reasoning models (audio reasoning tasks, See A4), validating its robustness and general applicability.
>
> Our adaptive mechanism $\mathcal{L}_{RL} + (\beta_0 - A) \cdot \mathcal{L}_D$ as formulated in Equation (4) provides intuitive understanding for exploration-exploitation balance: positive advantages (good samples) reduce regularization to encourage exploitation, while negative advantages (poor samples) increase regularization to maintain stability. The advantage clipping ensures bounded adaptation, preventing instability while maintaining meaningful regularization adjustment.
>
> We agree that more theoretical analysis is an area where more exciting and novel work can be made, and we would be interested in exploring the theoretical properties of dynamic regularization, particularly convergence guarantees and optimality conditions, in future work following your suggestion.
>
> > **Q3: I would suggest to add more explanation of advantage function estimation in particular for the section 3.4. ADRPO for Fine-tuning LLMs. At least summarise main ideas from the paper [29].**
>
> **A3**: Thank you for the constructive suggestion. We would like to clarify that paper [29] discusses Generalized Advantage Estimation (GAE), which normally relies on a critic model to estimate value functions. In contrast, in Section 3.4, we adopt the group-relative advantage estimation method from GRPO, which is fundamentally different from GAE. Specifically, GRPO does not require a critic model and instead computes advantages directly from outcome rewards by comparing samples within a batch or group. This approach is simpler and more stable, making it particularly well-suited for fine-tuning LLMs where training a separate critic can be challenging and computationally inefficient. We included the description of advantage estimation for the flow-matching version in Section 3.3, which follows similar principles to the LLM version.
>
> Following your suggestion, we will include more detailed descriptions of our advantage estimation in the final revision to enhance the clarity of our paper.
>
> > **Q4: I would also suggest to add more experiments replacing Qwen by other comparable models.**
>
> **A4**: We appreciate this insightful suggestion. In addition to Qwen models, **we have also provided extensive experiments on SD3 models for text-to-image generation** (Table 1, Figures 1-3) and demonstrated ADRPO's effectiveness across both architectures. These two model architectures are fundamentally different - SD3 uses flow matching for continuous generation (e.g., image) while Qwen uses autoregressive generation for discrete sequences (e.g., text) - yet both achieve significant improvements over baseline methods, sufficiently demonstrating the universal applicability of our approach. Besides, we chose Qwen models because they are widely used in RL post-training and represent state-of-the-art performance across various tasks [2-4], making them ideal choices for validating our method's effectiveness.
>
> To further address the concern, we conducted additional experiments fine-tuning the multi-modal reasoning model Qwen2.5-Omni-7B [5] (bf16 precision) on the AVQA dataset [6] with verifiable rewards and format rewards, evaluated on the MMAU benchmark [7]. Our results demonstrate that ADRPO significantly outperforms both baseline RL methods and strong commercial models:
>
> | Method | Sound (%) | Music (%) | Speech (%) | Total Accuracy (%) |
> |:--------:|:---------:|:---------:|:----------:|:------------------:|
> | Qwen2.5-Omni-7B (base model) | 72.37 | 64.37 | 69.07 | 68.6 |
> | **ADRPO (Ours)** | **81.98** | **70.06** | **75.98** | **76.0** |
> | GRPO (baseline RL) | 77.18 | 70.66 | 74.77 | 74.2 |
> | Gemini 2.5 Pro | 75.08 | 68.26 | 71.47 | 71.6 |
> | GPT-4o Audio | 64.56 | 56.29 | 66.67 | 62.5 |
>
> This demonstrates that our adaptive regularization approach achieves consistent improvements across three distinct model families: **flow matching models** (SD3), **autoregressive LLMs** (Qwen2/3), and **multi-modal reasoning models** (Qwen2.5-Omni), confirming the broad applicability across diverse architectures and modalities.
>
> > **Q5: Is it possible to overcome the problem of choosing baseline parameter $β_0$?**
>
> **A5**: Thank you for this important question. Our method actually simplifies the hyperparameter selection process compared to existing approaches. Rather than requiring extensive tuning of $β_0$, we demonstrate that ADRPO works effectively by simply adopting the same  values used in existing fixed regularization methods. For example, in our experiments (Section 4.1), we set $β_0=0.04$ for LLM fine-tuning (matching GRPO's fixed $β$) and $β_0=1$ for SD3 fine-tuning (matching baseline flow matching methods), without any additional hyperparameter search.
>
> This design choice serves two purposes: first, it ensures fair comparison by using identical baseline regularization strengths; second, it makes ADRPO a plug-and-play enhancement to existing methods. Users can directly replace fixed regularization with our adaptive approach using their current fixed regularization weight as $β_0$, immediately benefiting from superior exploration-exploitation trade-offs without additional tuning overhead.
>
> The substantial performance improvements we observe across all tasks—despite using the same  $β_0$ values as baseline methods—demonstrate that our core contribution lies in the adaptive mechanism itself rather than in hyperparameter optimization. Our advantage-based regularization adaptation automatically handles the exploration-exploitation balance that fixed methods cannot achieve.
>
>
> [1] Alonso, Noguer I. "The Mathematics of Group Relative Policy Optimization: A Multi-Agent Reinforcement Learning Approach." 2025.
>
> [2] Guo, Daya, et al. "Deepseek-r1: Incentivizing reasoning capability in llms via reinforcement learning." 2025.
>
> [3] M. Chen, et al. ReSearch: Learning to Reason with Search for LLMs via Reinforcement Learning. 2025.
>
> [4] Y. Fu, et al. RLAE: Reinforcement Learning-Assisted Ensemble for LLMs. 2025.
>
> [5] Xu, Jin, et al. "Qwen2.5-omni technical report." 2025.
>
> [6] Yang, Pinci, et al. "Avqa: A dataset for audio-visual question answering on videos." ACM MM. 2022.
>
> [7] Sakshi, S., et al. "Mmau: A massive multi-task audio understanding and reasoning benchmark." 2024.

---

> > ### Comment · Reviewer_4P4K · 2025-08-04
> > **discussion**
> >
> > I thank the authors for their answers. I'll keep my score.

---

> > > ### Author Response · Authors · 2025-08-04
> > > **Thank you**
> > >
> > > Thank you for your thoughtful review and for carefully considering our responses. We are grateful for your constructive feedback throughout the review process, which has further enhanced our work's presentation and clarity.

---

### Official Review · Reviewer_Kr8c · 2025-07-01

**Clarity:** 3
**Significance:** 3
**Originality:** 3
**Rating:** 5
**Confidence:** 3

**Summary:**

The paper introduces Adaptive Divergence Regularized Policy Optimization (ADRPO), a method for reinforcement learning fine-tuning of generative models with adaptive regularization, allowing for balanced exploitation-exploration tradeoffs, learning from useful examples while not being destructive to generative model capabilities.

The method replaces traditional fixed-weight W2 or KL-divergence regularization with one weighted dynamically with an Advantage estimate for the current sample. The technique is applied to both RL fine-tuning of flow matching, and RL fine-tuning of LLMs, beating non-adaptive baselines on both, with both quantitative and qualitative metrics.

The exploration-exploitation tradeoff is also quantified with measures of sample diversity and model divergence, demonstrating that the ADRPO technique leads to an optimal combination of both diversity, low model divergence and final reward.

**Questions:**

* Regarding small improvements: Did you produce a breakdown of whether it is possible for the baselines to achieve comparable results with more compute expenditure? And if so, with how much more expenditure?
* Could you provide more details on how does the procedure behave when the regularization coefficient is allowed to be negative vs when it is clipped?

**Ethical Concerns:**

["NO or VERY MINOR ethics concerns only"]

**Final Justification:**

The authors addressed my concerns in the rebuttal, I thus raised my score to a 5.

**Limitations:**

yes

**Quality:**

3

**Strengths And Weaknesses:**

Strengths:
* The proposed technique is very simple, but reliably improves the performance of RL finetuning techniques on the exploration-exploitation tradeoff frontier.
* The approach is tested on both continuous generative modeling, for flow matching based image generation, and discrete generative modeling, in LLM finetuning, showcasing that the underlying insight is not model or modality dependent, but is generally applicable to generative models.

Weaknesses:
* Quantitative and qualitative comparisons do not straightforwardly appear as a fundamental improvement over the state of the art: benchmark improvements are relatively small, and image generation quality does not seem to conclusively be much beyond other methods.

---

> ### Author Rebuttal · Authors · 2025-07-30
>
> Thank you for the thoughtful and constructive review. Your feedback has been invaluable in strengthening our paper's clarity and impact. We greatly appreciate the recognition of our method's simplicity and reliability in improving exploration-exploitation tradeoffs, as well as its general applicability across both continuous and discrete generative modeling tasks. We appreciate the recognition of our approach's ability to achieve balanced exploitation-exploration tradeoffs while preserving generative model capabilities, and that our method consistently outperforms non-adaptive baselines across both flow matching and LLM fine-tuning with superior quantitative and qualitative metrics. Below, we address each of your concerns systematically and will incorporate these clarifications and analysis into our final revision.
>
> > **Q1: Quantitative and qualitative comparisons do not straightforwardly appear as a fundamental improvement over the state of the art: benchmark improvements are relatively small, and image generation quality does not seem to conclusively be much beyond other methods.**
>
> **A1**: Thank you for this insightful comment. We would like to clarify several key aspects of our significant improvements:
>
> **Quantitative Improvement**: As shown in Table 1, when using the same base model (SD3), our benchmark performance on preference scores (ClipScore, BLIPScore, ImageReward, PicScore), Image Quality and Diversity consistently surpasses all other RL post-training methods. More remarkably, compared to much larger models like FLUX.1-Dev (12B) and SANA-1.5 (4.8B), our significantly smaller 2B model outperforms them across all evaluation metrics. These improvements are substantial because we achieve gains across all dimensions simultaneously rather than sacrificing one metric for another, avoiding the trade-offs that plague existing methods.
>
> **Qualitative Improvement**: The visual evidence in Figures 1 and 5 demonstrates clear semantic alignment and generation quality improvement. In Figure 1, for complex compositional tasks like "A green apple and a black backpack," FLUX.1-Dev (12B) incorrectly generates an apple larger than the backpack, while the SD3 base model misunderstands the quantity, generating two apples instead of one. SANA-1.5 (4.8B) produces common sense errors like generating an astronaut with three arms for "Van Gogh style astronaut." Particularly evident in Figure 5, the spatial relationship task "A zebra to the right of a fire hydrant" reveals systematic failures across all baseline models: FLUX.1-Dev (12B) completely fails to generate the zebra, SANA-1.5 (4.8B) misplaces the spatial relationship with the zebra (6 legs) positioned incorrectly, while the SD3 base model generates the zebra and fire hydrant but fails to establish the correct positional relationship. Our method correctly handles these challenging cases involving attribute binding, spatial relationships, and compositional reasoning.
>
> **RL Method Comparison**: In Figures 2 and 6, baseline RL methods consistently demonstrate failure patterns. DPO maintains diversity but demonstrates poor semantic alignment - generating zebras with unnatural images (extra legs), failing text rendering ("Diffusion" storefronts), and incorrect object quantities ("A green apple and a black backpack" produces multiple apples in Figure 6). ORW-CFM-W2 exhibits similar failures with malformed zebras, text rendering issues, and compositional errors (e.g., an apple as large as the backpack in Figure 6). Our ADRPO method successfully avoids these fundamental errors across all tasks, demonstrating superior control over naturalness, text rendering, object counting, positioning, and color attribution while maintaining generation diversity.
>
> **Additional Evidence of Substantial Improvements in Multi-Modal Reasoning Tasks**: To further validate our method's effectiveness, we conducted additional experiments fine-tuning the multi-modal reasoning model Qwen2.5-Omni-7B [1] (bf16 precision) on the AVQA dataset [2] with verifiable rewards and format rewards, evaluated on the MMAU benchmark [3]. On the MMAU benchmark [3], ADRPO significantly outperforms both baseline RL methods and strong commercial models:
>
> | Method | Sound (%) | Music (%) | Speech (%) | Total Accuracy (%) |
> |:--------:|:---------:|:---------:|:----------:|:------------------:|
> | Qwen2.5-Omni-7B (base model) | 72.37 | 64.37 | 69.07 | 68.6 |
> | **ADRPO (Ours)** | **81.98** | **70.06** | **75.98** | **76.0** |
> | GRPO (baseline RL) | 77.18 | 70.66 | 74.77 | 74.2 |
> | Gemini 2.5 Pro | 75.08 | 68.26 | 71.47 | 71.6 |
> | GPT-4o Audio | 64.56 | 56.29 | 66.67 | 62.5 |
>
> Beyond quantitative improvements, the qualitative analysis reveals superior reasoning capabilities. For instance, in complex audio reasoning tasks (more results will be included in our final revision):
>
> **Sample ID:** ff7bff97-342e-4285-bbb9-15841364b072
> - **Question**: "Based on the given audio, identify the source of the flowing water. Choices: Bathtub, River, Fountain, Rain"
> - **ADRPO**: "The audio suggests a continuous, steady flow of water, which is characteristic of a bathtub. The other options (river, fountain, rain) typically have more varied or intermittent sounds." → **Bathtub** ✓
> - **GRPO**: "The sound is consistent with a steady flow of water, likely from a man-made source like a bathtub or fountain rather than natural sources like a river or rain." → **Fountain** ✗
>
>
> **Sample ID:** b7701ab1-c37e-49f2-8ad9-7177fe0465e9
> - **Question**: "What was the last sound in the sequence? Choices: footsteps, dog_barking, camera_shutter_clicking, tapping_on_glass"
> - **ADRPO**: "The sequence of sounds includes footsteps, a dog barking, a camera shutter clicking, and a tapping on glass. The last sound is a distinct 'tapping on glass', which is a clear and unique sound compared to the others." → **tapping_on_glass** ✓
> - **GRPO**: "The last sound in the sequence is a camera shutter click, which is a distinct sound from the other options provided." → **camera_shutter_clicking** ✗
>
>
> This demonstrates that our adaptive regularization mechanism enables superior step-by-step reasoning and reduces failure modes compared to fixed regularization RL methods across diverse modalities, confirming the broad applicability and effectiveness of our approach beyond visual generation tasks.
>
>
> > **Q2: Regarding small improvements: Did you produce a breakdown of whether it is possible for the baselines to achieve comparable results with more compute expenditure? And if so, with how much more expenditure?**
>
> **A2:** Thank you for this important question. As demonstrated in our learning curves in Figures 7, 8, and 9, the improvements from **our ADRPO are not achievable through additional compute** for baseline methods because our method fundamentally changes the learning dynamics, exploration-exploitation trade-off and final convergence points.
>
> In Figure 7, both DPO and ORW-CFM-W2 exhibit plateau behavior in their reward curves during training, indicating they have converged to stable but suboptimal solutions. Even with further increases in training time, these methods cannot reach ADRPO's performance levels due to their inherently limited exploration-exploitation trade-offs. **The final convergence points clearly demonstrate that our method achieves a superior reward-diversity Pareto frontier compared to baseline methods.**
>
> More critically, in Figures 8 and 9 for LLM fine-tuning, baseline methods like GRPO actually suffer from model training collapse during extended training, where performance degrades rather than improves with longer training time. This demonstrates that additional compute is not only ineffective but potentially harmful for fixed regularization methods. In contrast, ADRPO's adaptive regularization mechanism enables it to discover better regions in the solution space that fixed methods cannot reach regardless of training duration (See Figure 4), fundamentally achieving better exploration-exploitation trade-off rather than simply requiring more computational resources.
>
>
> > **Q3: Could you provide more details on how does the procedure behave when the regularization coefficient is allowed to be negative vs when it is clipped?**
>
> **A3**: Thank you for this insightful suggestion. Our main contribution is introducing advantage-based adaptive regularization, which has demonstrated substantial improvements over fixed regularization across various tasks in our paper.
>
> We use advantage clipping to ensure $β_{tot} = β_0 - A$ remains positive and stable. To address your concern about clipping behavior, we provide ablation studies on fine-tuning Qwen2.5-Omni-7B with different clipping ranges:
>
> | Clipping Range | Amin | Amax | Sound (%) | Music (%) | Speech (%) | Total Accuracy (%) |
> |:----------------:|:------:|:------:|:-----------:|:-----------:|:------------:|:-------------------:|
> | 1×β0 (recommended) | -0.04 | 0.04 | 81.98 | 70.06 | 75.98 | **76.0** |
> | 2×β0 | -0.08 | 0.08 | 84.08 | 69.46 | 73.57 | 75.7 |
> | 0.5×β0 | -0.02 | 0.02 | 82.58 | 71.26 | 74.47 | 76.1 |
> | GRPO (baseline) | - | - | 77.18 | 70.66 | 74.77 | 74.2 |
>
> **When allowing negative regularization** (2×β0 setting), the coefficient can become negative ($β_{tot} < 0$), reversing the regularization effect and encouraging greedy exploitation. While this achieves higher performance on specific tasks (84.08% on sound), it reduces generalization across other modalities (73.57% vs. 75.98% on speech). This demonstrates that excessive negative regularization leads to aggressive optimization at the expense of generalization. We recommend maintaining positive coefficients through appropriate clipping (1×β0) for better balance and more robust performance.
>
> [1] Xu, Jin, et al. "Qwen2.5-omni technical report." 2025.
>
> [2] Yang, Pinci, et al. "Avqa: A dataset for audio-visual question answering on videos." 2022.
>
> [3] Sakshi, S., et al. "Mmau: A massive multi-task audio understanding and reasoning benchmark." 2024.

---

> > ### Comment · Reviewer_Kr8c · 2025-08-06
> >
> > Thank you for the detailed rebuttal, I will raise my score to a 5.

---

> > > ### Author Response · Authors · 2025-08-06
> > > **Thank you**
> > >
> > > Thank you for taking the time to carefully consider our rebuttal. We are grateful for the improved score. We appreciate your constructive feedback throughout the review process, which has been invaluable in strengthening our paper's clarity and impact.

---

### Note · Authors · 2025-08-14

We sincerely thank all reviewers and Area Chairs for their time and effort throughout the review process. We greatly appreciate that our advantage-based adaptive regularization mechanism, motivation for balancing exploration-exploitation trade-offs, convincing experimental results, and broad applicability to both continuous and discrete generative models have been acknowledged by the reviewers.

We are particularly grateful for the reviewers' unanimous positive support of our work. The thorough evaluations and constructive feedback from the reviewers have significantly enhanced the technical rigor and clarity of our work. All clarifications, additional experimental results (including the new multi-modal reasoning experiments on Qwen2.5-Omni-7B), and detailed analyses provided in our rebuttals will be comprehensively integrated into our final revision. We thank the reviewers for their dedication to responsible reviewing and for helping us improve our paper.

---

### Decision · Program_Chairs · 2025-09-17

**Decision:**

Accept (poster)

**Comment:**

The authors propose an adaptive regularization approach based on advantage functions for fine-tuning generative models, including LLMs and diffusion models. Many reviewers agree that the paper demonstrates strong empirical performance. However, reviewers raised concerns about whether the proposed method is truly well-founded, and the overhead to estimate the advantages functions.